# A Practical Investigation of Spatially-Controlled Image Generation with Transformers

**Guoxuan Xia, Harleen Hanspal, Petru-Daniel Tudosiu,**
**Shifeng Zhang & Sarah Parisot**
*Work done at Huawei Noah's Ark Lab*

*g.xia21@imperial.ac.uk*
*zhangshifeng4@huawei.com*

**Reviewed on OpenReview:** *https://openreview.net/forum?id=loT6xhgLYK*

## Abstract

Enabling image generation models to be *spatially controlled* is an important area of research, empowering users to better generate images according to their own fine-grained specifications via *e.g.* edge maps, poses. Although this task has seen impressive improvements in recent times, a focus on rapidly producing stronger models has come at the cost of detailed and fair scientific comparison. Differing training data, model architectures and generation paradigms make it difficult to disentangle the factors contributing to performance. Meanwhile, the motivations and nuances of certain approaches become lost in the literature. In this work, we aim to provide clear takeaways *across* generation paradigms for practitioners wishing to develop *transformer*-based systems for spatially-controlled generation, clarifying the literature and addressing knowledge gaps. We perform *controlled experiments* on ImageNet across diffusion-based/flow-based and autoregressive (AR) models. First, we establish control token prefilling as a simple, general and performant baseline approach for transformers. We then investigate previously underexplored *sampling time* enhancements, showing that extending classifier-free guidance to control, as well as softmax truncation, have a strong impact on control-generation consistency. Finally, we re-clarify the motivation of adapter-based approaches, demonstrating that they mitigate "forgetting" and maintain generation quality when trained on limited downstream data, but underperform full training in terms of generation-control consistency. Code: *https://github.com/guoxoug/transformer-imagenet-ctrl*.

## 1 Introduction

In recent years, deep-learning-based image generation has advanced at an unprecedented pace, finding widespread adoption in many applications. This has been driven by diffusion-based (Song et al., 2021; Lipman et al., 2023), and more recently, autoregressive (AR) (Tian et al., 2024; Sun et al., 2024) and masked (Chang et al., 2022; Yu et al., 2024) modelling approaches that provide a stable and scalable paradigm for learning complex and diverse image distributions. Additionally, *latent space* modelling (Esser et al., 2021; Rombach et al., 2022) has facilitated efficient, high-resolution generative modelling. Notably, although earlier approaches for large-scale image generation were based on diffusion with the UNet architecture (Ronneberger et al., 2015; Rombach et al., 2022; Podell et al., 2024), the state-of-the-art is now dominated by *transformers* (Vaswani et al., 2017), thanks to their scalability (Ma et al., 2024; Tian et al., 2024; Esser et al., 2024; Labs, 2024). Users most commonly control the content of generated images via text conditioning; however, this lacks spatial precision due to the inherent ambiguity of text and models' limited text understanding (Huang et al., 2023). This has led to increasing interest in *spatially-conditioned* image generation, especially for creative applications, where the user can *control* the fine-grained structure of generations via an input such as an edge or depth map (Zhang et al., 2023; Mou et al., 2024). In this work, we aim to *clarify* design choices for the above task for the latest (transformer-based) generation paradigms, with the objective of providing practitioners with useful and practical takeaways. To this end, we identify a number of **gaps in the research literature**, which may encumber a practitioner wishing to build their own system:

1) **Unclear comparisons**: Existing work on spatially-conditioned image generation tends to focus on rapidly producing high-performing models that can be open-sourced and contribute to the research and general practitioner communities (Xiao et al., 2025; Tan et al., 2025a; Li et al., 2025; Zhang et al., 2025b). Although this advances the field, it comes at the cost of scientific clarity/understanding. Approaches with different training data (pre-training and control finetuning),

different architectures and different generation paradigms are compared against each other, limiting useful insight into why each approach may be more or less effective for practitioners. 2) **Sampling enhancements**: Research has focused on new architectures (Zhang et al., 2023; Tan et al., 2025a; Li et al., 2025), and new training losses (Li et al., 2024a; Zhang et al., 2025a) but has neglected *generation-time* algorithmic adjustments, even though such approaches may be simple to implement and offer low-cost post-training improvements. 3) **Adapters vs full training**: There is a wide range of approaches to the problem, with differing *design motivations* and *architectural choices*, without clear positioning and comparison. In particular, the choice between fully training all parameters versus freezing the weights of a pretrained model and only training an adapter (Zhang et al., 2023) is often poorly presented or ignored.

Motivated by the above, this work aims to provide a clear comparative investigation for spatially-conditioned image generation, with practical takeaways *across* transformer-based architectures and generation paradigms. To this end, we perform *controlled experiments* on ImageNet (Deng et al., 2009) over two representative but contrasting generative modelling approaches: Scalable Interpolant Transformer (SiT) (Ma et al., 2024) and Visual Autoregressive Modelling (VAR) (Tian et al., 2024). The former represents widespread diffusion/flow-based approaches, the latter recently appearing but under-investigated LLM-inspired autoregressive approaches. Our **key takeaways** are summarised as follows:

1. Out of the plethora of possible approaches, we argue that transformer prefilling with control tokens is an obvious starting point for architectural design. We demonstrate empirically that this simple and general approach is a strong baseline that performs well out of the box across different generation paradigms.
2. We investigate simple sampling enhancements, finding some to have *meaningful practical impact*. Extending classifier-free guidance (CFG) to spatial control greatly improves generation-control consistency but trades off quality and inference cost, whilst softmax truncation improves *both* image quality *and* control consistency for VAR.
3. We re-clarify the motivation of control-adapters, where a pretrained generative model is *frozen* and an adapter module is trained – to *preserve* generation ability and mitigate "forgetting" when training on limited/undiverse downstream data. We empirically demonstrate this property, but also find that adapters consistently underperform prefill + fully training all parameters for generation-control consistency when downstream data is not limited.

## 2 Preliminaries

**Image generation with spatial control.** This is a task popularised by ControlNet (Zhang et al., 2023), where the user wants to generate an output that adheres spatially to a supplied conditioning image, *e.g.* an edge, depth or segmentation map. This enables greater user control over generations, which is important for creative applications *e.g.* a user may wish to generate a person in a specific pose. Formally, we generate samples $x$, given control condition $c$ and prompt or class conditioning $y$ using a model of the conditional distribution over $x$ parameterised by $\theta$,

$$x \sim p_\theta(x|y, c) . \tag{1}$$

**Latent image modelling.** For high-resolution generation, models are typically broken into two stages: an autoencoder, which maps images to a spatially compressed lower-resolution latent space, and a generative model that operates in this latent space (van den Oord et al., 2017; Esser et al., 2021; Rombach et al., 2022). Since image *semantics* are still preserved at lower resolutions, greater computational efficiency can be achieved by performing generative modelling in the latent space (Chen et al., 2025a). Formally, the autoencoder is made up of an encoder $\mathcal{E}$ that compresses image $x$ to latent embedding $z$ and a decoder $\mathcal{D}$ that aims to reconstruct the image from the latent. Generation is then performed in the *latent* space, with sampled latents then decoded into images using $\mathcal{D}$,

$$z = \mathcal{E}(x), \quad \hat{x} = \mathcal{D}(z), \quad z \sim p_\theta(z|y, c) . \tag{2}$$

From this point onwards, we assume that all generative modelling is w.r.t. $z$ in the latent space.

**Diffusion-based image generation.** The most popular approach for image generation currently is diffusion (or flow)-based models (Song et al., 2021; Lipman et al., 2023). These models learn to denoise data, generating samples by starting from pure noise and iteratively denoising to a clean sample. The *forward process*, over time $t \in [0, 1]$, is defined by,

$$z_t = \alpha_t z_0 + \sigma_t \epsilon, \quad \epsilon \sim \mathcal{N}(0, I), \tag{3}$$

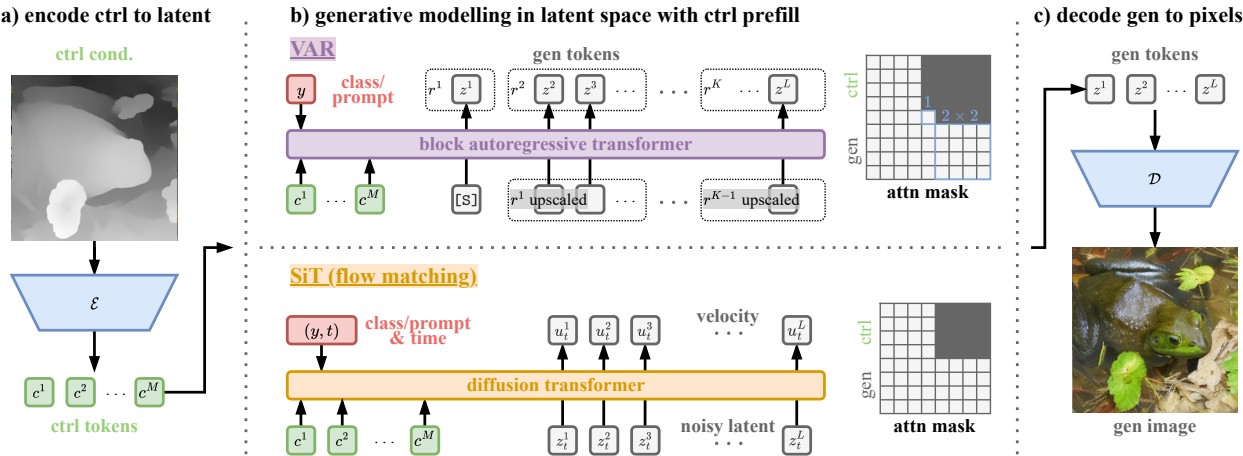

Figure 1: Illustration of our *prefill* baseline for spatial control with transformers. **a)** The control conditioning is encoded to the generative model's latent space. **b)** Generative modelling is performed in the latent space using VAR (top) or SiT (bottom). Generative tokens *attend back* to conditioning control tokens using a *block casual* mask, allowing for KV-caching at inference. **c)** Generated tokens are decoded from the latent space to the pixel space.

where noise $\epsilon$ is mixed with the clean sample $z_0$ according to the *noise schedule* $(\alpha_t, \sigma_t)$. Clean data $z_0$ can then be generated from noise $z_1$ by integrating the following *probability flow ODE*[1] (Song et al., 2021),

$$\underbrace{u_t = \frac{dz_t}{dt}}_{\text{flow vector}} = f_t z_t - \frac{1}{2} g_t^2 \underbrace{\nabla_{z_t} \log p(z_t)}_{\text{score } s(z_t)}, \quad z_1 \sim \mathcal{N}(z_1; \mathbf{0}, \mathbf{I}), \qquad \text{where } f_t = \frac{da_t}{dt}\frac{1}{a_t}, \quad g_t^2 = \frac{d\sigma_t^2}{dt} - 2f_t \sigma_t^2. \quad (4)$$

Diffusion models are trained to model various parameterisations of the score $s_\theta(z_t, t) \approx \nabla_{z_t} \log p(z_t)$, *e.g.* data ($z_0$) or noise ($\epsilon$) prediction $s_\theta(z_t, t) = [\alpha_t z_\theta(z_t, t) - z_t]/\sigma_t^2 = -\epsilon_\theta(z_t, t)/\sigma_t$, whilst flow matching directly models the probability flow $u_\theta(z_t, t) \approx u_t$. We note that $u, s, \epsilon$ can be directly calculated from each other given $z_t$ (Gao et al., 2024).

**Visual Autoregressive modelling (VAR).** Recently, inspired by the success of large language models, interest has grown in developing autoregressive-style approaches for image generation (Esser et al., 2021; Sun et al., 2024). In particular, Visual Autoregressive modelling (VAR) (Tian et al., 2024), stands out as a strong performer, rivalling the generation quality of diffusion-based approaches. The core idea is to model image generation as a course-to-fine progression of "scales", *i.e.* image representations of increasing spatial resolution, allowing a transformer-based model to progressively generate an image conditioned on lower-resolution representations via "next-scale prediction". This image-specific approach outperforms existing raster-scan AR models. The joint distribution over scales is modelled as

$$p_\theta(r^1, r^2, \ldots, r^K) = \prod_{k=1}^{K} p_\theta(r^k \mid r^1, r^2, \ldots, r^{k-1}) \quad \text{for image scales } \{r^1, r^2, \ldots, r^K\}, \quad (5)$$

allowing each scale of $h^k \times w^k$ tokens to be generated given the sequence of previously generated scales (within-scale tokens are generated in parallel). VAR discretises the latent space using a vector-quantised variational autoencoder (VQVAE) (van den Oord et al., 2017), allowing for modelling using softmax and training via cross entropy.[2]

**Classifier-free guidance (CFG).** A standard method for improving generation quality is classifier-free guidance (Ho & Salimans, 2021), which works by adjusting the generative distribution so that probability mass/density is concentrated in regions where the probability of the prompt/class $p(y|z)$ is implicitly higher,

$$p_\theta^{\text{guid}}(z) = \frac{1}{Z} p_\theta(z|y) \left[\frac{p_\theta(z|y)}{p_\theta(z)}\right]^{\gamma_y - 1}, \quad \text{since } p(y|z) \propto \frac{p(z|y)}{p(z)}, \quad (6)$$

---

[1]We omit stochastic samplers/SDE solvers (Song et al., 2021) for brevity, as we do not use them in this work.

[2]We omit some of the details with respect to the multi-scale and residual nature of VAR's VQVAE (Tian et al., 2024) for brevity. We also note that there have been a number of iterations on VAR (Ren et al., 2025; Jiao et al., 2025), however, we stick to evaluating the original in this work.

where $\gamma$ (typically $\geq 1$) is the guidance scale/strength and $Z$ is a normalising constant. To achieve this typically a model is trained to model both $p_{\boldsymbol{\theta}}(\boldsymbol{z}|\boldsymbol{y})$ and $p_{\boldsymbol{\theta}}(\boldsymbol{z})$ via an empty condition $\boldsymbol{y} = \varnothing$. Applying guidance to diffusion and VAR,

$$\boldsymbol{s}_{\boldsymbol{\theta}}^{\text{guid}}(\boldsymbol{z}_t|\boldsymbol{y}) = \boldsymbol{s}_{\boldsymbol{\theta}}(\boldsymbol{z}_t|\boldsymbol{y}) + (\gamma_y - 1)\left[\boldsymbol{s}_{\boldsymbol{\theta}}(\boldsymbol{z}_t|\boldsymbol{y}) - \boldsymbol{s}_{\boldsymbol{\theta}}(\boldsymbol{z}_t)\right], \quad \boldsymbol{v}_{\boldsymbol{\theta}}^{\text{guid}}(\boldsymbol{r}|\boldsymbol{y}) = \boldsymbol{v}_{\boldsymbol{\theta}}(\boldsymbol{r}|\boldsymbol{y}) + (\gamma_y - 1)\left[\boldsymbol{v}_{\boldsymbol{\theta}}(\boldsymbol{r}|\boldsymbol{y}) - \boldsymbol{v}_{\boldsymbol{\theta}}(\boldsymbol{r})\right], \quad (7)$$

results in simple linear combinations of the score estimates $\boldsymbol{s}_{\boldsymbol{\theta}}(\boldsymbol{z}_t, t) \approx \nabla_{\boldsymbol{z}_t} \log\ p(\boldsymbol{z}_t)$ (see Eq. (4)) and pre-softmax logits $\boldsymbol{v}$. Note that the normalising constant $Z$ is naturally dealt with for diffusion due to the score being a gradient, whilst in VAR it is taken care of by the softmax denominator. Note that setting $\gamma_y = 0$ recovers the unconditional model $p_{\boldsymbol{\theta}}(\boldsymbol{z})$, whilst $\gamma_y = 1$ recovers the conditional one $p_{\boldsymbol{\theta}}(\boldsymbol{z}|\boldsymbol{y})$. $\gamma_y$ is typically set to be $\geq 1$ in order to improve generation quality.

**Related work in spatially controlled image generation.** Earlier approaches to this problem involve freezing the UNet (Ronneberger et al., 2015) of Stable Diffusion (Rombach et al., 2022) and training an additional "adapter" module that injected control information via attention (Gligen (Li et al., 2023)), or addition, (ControlNet (Zhang et al., 2023), T2I-Adapter (Mou et al., 2024)). Later research focuses on improving ControlNet by unifying controls into a single model (Qin et al., 2023; Zhao et al., 2023) and refining the training loss to optimise control-generation consistency (Li et al., 2024a; Zhang et al., 2025a). As image generation models diversified from UNets to transformers (Vaswani et al., 2017), and from diffusion to AR (Tian et al., 2024; Sun et al., 2024) and masked generation (Chang et al., 2022), a wider range of architectural approaches have arisen. Adapters for diffusion/flow transformers include Pixart-$\delta$ (Chen et al., 2024a) which adapts ControlNet for Pixart-$\alpha$ (Chen et al., 2024b) and OminiControl (Tan et al., 2025a;b), which explores attention adapters for Flux (Labs, 2024). ControlAR (Li et al., 2025) adds spatial control by fully fine-tuning a raster-scan AR model with an addition-based control encoder, whilst ControlVAR (Li et al., 2024b) proposes to jointly model image and control data using a VAR model. Recently, there has been a surge in research aiming to fully train unified transformer foundation models that can perform generation and understanding over various tasks and modalities (Zhang et al., 2025b). Notably, Omnigen (Xiao et al., 2025) and UniReal (Chen et al., 2025b) train spatially-conditioned diffusion transformers where the conditions are simply processed as additional tokens. This wide array of diverse algorithms and architectures in the literature may seem daunting to newcomers. This work aims to provide some clear and practical takeaways to practitioners wanting to use transformers for spatially controlled image generation.

## 3 Evaluation Setup

**Data.** In order to perform a clean and fair comparison and avoid potential confounding factors related to data quality (during both pre-training and control finetuning), we train and evaluate on class-conditioned ImageNet (Deng et al., 2009), a well-established benchmark for image generation. ImageNet is at a scale ($\sim$1.2M samples, 256$\times$256 resolution, baseline models with $10^8 \sim 10^9$ parameters) that is close to deployed real-world image generation models, whilst still having manageable training and evaluation costs for an academic compute budget. We train and evaluate at a resolution of 256$\times$256 over two distinct spatial control conditions, which we extract from the ImageNet dataset on the fly during training and evaluation: 1) Canny edge maps (Canny, 1986), using `kornia` with fixed thresholds $(0.1, 0.2)$, and 2) dense depth maps using DPT-Hybrid (Ranftl et al., 2021).

**Evaluating generated images.** For most evaluations we generate 10K samples for evaluation, conditioned on controls extracted from the first 10 images of each of the 1000 classes in the ImageNet validation dataset. In a few cases, to compare with the literature, we generate using controls from all 50K validation images. We use fixed random seeds. We measure image generation performance using three metrics. Fréchet Inception Distance (FID) (Salimans et al., 2016) measures the distributional distance between the data distribution and generated samples, capturing both realism and diversity, whilst Inception Score (IS) (Heusel et al., 2017) measures the (class-)diversity and clarity of generations, reflecting how confidently a pretrained classifier can assign labels and how varied those labels are across samples. These are widely used image quality metrics. Generation-control consistency (Li et al., 2024a) intuitively measures how closely generations adhere to spatial control, via similarity between the original control and the control extracted from the generated image. Following Li et al. (2024a), for Canny edge maps we compare binary pixels using F1 score $\uparrow$, whilst for depth maps we use root mean square error $\downarrow$ (RMSE) for pixel values in $[0, 255]$. Consistency is a naturally desirable property, *i.e.* to have the generation closely match the provided conditioning, and is especially motivated by applications with tight spatial tolerances such as industrial design. Additionally we emphasise the following:

> **Takeaway:** Metrics for generation quality (FID, IS) and control consistency (F1, RMSE) quantify certain aspects of performance; however, practitioners' requirements subjectively depend on their individual use cases (*e.g.* prioritise consistency vs aesthetics, don't care about diversity). Readers should approach the quantitative results presented in this paper considering both the nature of the metrics reported and their own individual requirements.

**Evaluating inference cost.**   We compare generation throughput in images per second (img/s) and latency in milliseconds (ms) by directly measuring time in python. Inference is performed on a single NVIDIA Tesla V100-32GB-SXM2. We do not include the decoder to the pixel space, or the extraction and encoding of control data to the latent space. This is to isolate the impact on the transformer component of the model, as that is the design space we are exploring.

**Limitations.**   Although the aim of the above experimental setup is to provide practically useful and generalisable take-aways, we also qualify that it is not comprehensive. For example, other than the additional experiment in Appendix C, we do not perform experiments on text-to-image models. Neither do we directly evaluate raster-scan AR image generation models. Readers should take this into consideration when interpreting our results/contributions for their own use.

## 4   Prefilling – a Baseline for Transformer-Based Controllable Generation

In order to investigate design and deployment choices for image generation with spatial control for *transformer*-based (Vaswani et al., 2017) model architectures, we need to first establish a baseline approach. There are many potential options, an obvious choice being to adapt ControlNet from UNet to transformer as in Chen et al. (2024a). However, we argue that control token *prefilling* is a natural and simpler starting point. That is to say simply passing spatial control tokens $c^1, \ldots, c^M$ through the transformer stack, and then allowing generation-related tokens $z^1, \ldots, z^L$ to attend back to the control tokens (via block-causal attention mask during training and KV-cache at inference). Compared to other approaches (Chen et al., 2024a; Li et al., 2025) it is simpler, leaving the architecture untouched, and can be broadly applied to *any* transformer-based generation paradigm (diffusion, AR, *etc.*), rather than being *paradigm specific*. In addition, the use of a KV-cache means that inference is only performed once on the conditioning tokens, whereas other approaches may incur considerably more overhead running once for each generation step (Tan et al., 2025a;b). Finally, this simpler, unified single-transformer approach is of particular interest as it is increasingly popular in the literature (Zhang et al., 2025b), being applied beyond spatial control to multimodal generation, understanding and editing. Notably, the recent foundation model Omnigen (Xiao et al., 2025) adopts this prefilling approach for spatial control.

We investigate two recent transformer-based approaches for image generation, to cover both diffusion-based and (comparatively underexplored) autoregressive methods: 1) SiT (Ma et al., 2024), which trains a diffusion transformer (DiT) architecture (Peebles & Xie, 2023) using flow matching, and 2) VAR, which autoregressively generates image representations of increasing spatial scale (Tian et al., 2024). Fig. 1 illustrates our prefill baseline for both cases.

**Implementation and experimental details.**   To implement the above prefilling approach (Fig. 1), we simply fine-tune the existing architectures with additional conditioning tokens, which the generation tokens attend to via a block-causal mask. We consider VAR-d16 and SiT-XL/2, initialising both models with ImageNet-pretrained checkpoints.[3] The base architectures are left unchanged. For VAR we add new learnable positional embeddings for the control tokens and a shared additive "level" embedding for all control tokens, whilst for SiT we only add a new "level" embedding for the control tokens, reusing the original sinusoidal positional embeddings. For SiT, we set diffusion time $t = 0$ (clean data) for control tokens. We finetune for 10 epochs with control conditioning with batch size 256 (∼50K iterations) using the original optimisers and hyperparameters. Following the original papers we linearly increase guidance scale $\gamma_y$ from zero over generation scales for VAR, whilst keeping it constant for SiT. CFG is performed by only omitting class conditioning and not control conditioning. VAR and SiT use different image crops, leading to small differences in visualisations.

**Results.**   Fig. 2 shows results for our prefill baseline using default sampling parameters from the original papers.[45] We find that prefilling is able to achieve better image generation quality (FID↓) compared to existing transformer-based approaches (Li et al., 2024b; 2025) in the literature whilst achieving comparable control consistency (much better in the

---

[3]VAR: https://github.com/FoundationVision/VAR, SiT: https://github.com/sihyun-yu/REPA (Yu et al., 2025)
[4]Although we reduce the number of sampling steps for SiT to 64 to reduce evaluation time.
[5]We note that in the VAR paper CFG is defined with $\hat{\gamma} = \gamma + 1$, so in our convention the CFG scale is higher by 1 compared to Tian et al. (2024).

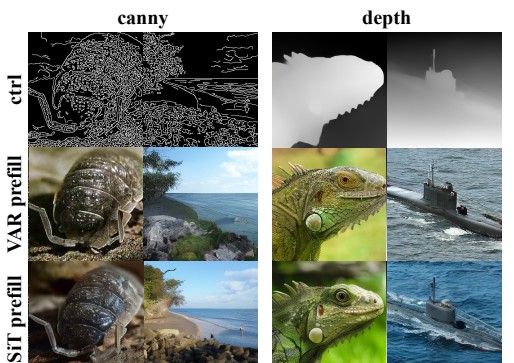

| model (sampling params) | control conditioning | #params | gen. quality | | | | ctrl consistency | | inf. cost | |
|---|---|---|---|---|---|---|---|---|---|---|
| | | | FID↓ 50K | IS↑ 50K | FID↓ 10K | IS↑ 10K | F1↑ | RMSE↓ | lat↓ (s) bs=1 | TP↑ (img/s) bs=16 |
| ImageNet-val (ctrl source) | | | 1.77 | 237 | 4.44 | 187 | | | | |
| VAR-d16 CFG=2.5, temp=1, top-p=0.96, top-k=900 | no ctrl | 310M | 3.36 | 277 | 5.71 | 214 | | | 0.26 | 17 |
| | canny prefill | | 3.94 | 190 | 6.60 | 158 | 34.6 | | 0.31 | 12 |
| | + ctrl quant | | 3.95 | 193 | 6.65 | 156 | 30.9 (d) | | | |
| | depth prefill | | 3.30 | 207 | 5.95 | 165 | | 31.3 | | |
| SiT-XL/2 CFG=1.5, Euler ODE, steps=64 | no ctrl | 675M | 2.01 | 300 | 4.47 | 228 | | | 3.0 | 0.39 (c) |
| | canny prefill | | 3.04 (a) | 205 | 5.69 | 161 | 39.3 | | 3.4 | 0.35 (c) |
| | depth prefill | | 2.78 | 218 | 5.43 | 170 | | 29.3 (b) | | |
| ControlVAR* VAR-d30 | canny | 2.0B | 7.85 | 162 | | | | | | |
| | depth | | 6.50 | 182 | | | | | | |
| ControlAR* LlamaGen-L | canny | 343M | 7.69 | | | | 34.9 | | | |
| | depth | | 4.19 | | | | | 31.1 | | |

Figure 2: **Left:** Examples of conditional generations using prefilling. Additional examples can be found in Appendix B. **Right:** Results for our prefill baseline where each model is finetuned from an ImageNet-pretrained base model. "*" indicates results copied from Li et al. (2025; 2024b). With default sampling parameters, prefilling is able to **(a)** generate higher quality images than recent transformer-based approaches with **(b)** comparable control consistency. We note the **(c)** relatively low overhead for SiT, as KV-caching means the control tokens use up only a single forward pass. **(d)** Quantising the control input with VQVAE hurts consistency for VAR, without affecting generation quality.

case of SiT). Notably our VAR-d16 considerably outperforms ControlVAR d-30, suggesting that the image-control joint modelling proposed by (Li et al., 2024b) is suboptimal for conditional generation. We note that control-conditioned IS is lower than for generations without control; however this is to be expected since the control inputs are sourced from the ImageNet validation dataset, which naturally has lower IS/more ambiguous samples. We also find that the use of KV-caching means that the latency and throughput overheads of prefilling compared to no-ctrl generation are quite reasonable, especially for SiT where the cache is calculated only once compared to the 64 denoising steps.

> **Takeaway:** Prefilling (Fig. 1) is a simple, general and effective baseline for spatial control with transformers.

We note that our result goes against the results in ControlAR (Li et al., 2025) that suggest that prefilling is a poor choice (they discard this design choice and opt for additive injection of control information, which is what we report in Fig. 2). We hypothesise that Li et al. (2025)'s choices to 1) vector *quantise* the control input, destroying information, and 2) enforce a triangular causal attention mask, limiting attention between control tokens, may have hurt performance. We perform our own experiment with VAR's VQVAE (Fig. 2) confirming the negative effect of quantising the control input. Although avoiding quantisation may sound obvious, AR models are often trained using token indices as inputs by default (Sun et al., 2024), meaning that "turning off" quantisation may lead to additional implementation complexity.

> **Takeaway:** Vector quantisation of the control input hurts generation-control consistency.

## 5   The Impact of Sampling Parameters

For our intial experiments we left sampling/inference hyperparameters untouched compared to the original VAR (Tian et al., 2024) and SiT (Ma et al., 2024) papers, however, is it natural to ask whether adjusting sampling parameters at generation time can provide improved performance "for free". To this end, we consider, classifer-free guidance (CFG), control guidance (ctrl-G) and sampling distribution truncation. We note that this aspect of generation has been surprisingly neglected in the literature, with research tending to focus on new architectures (Li et al., 2025) or improved training losses (Li et al., 2024a; Zhang et al., 2025a), often presenting evaluations with a single hyperparameter setting.

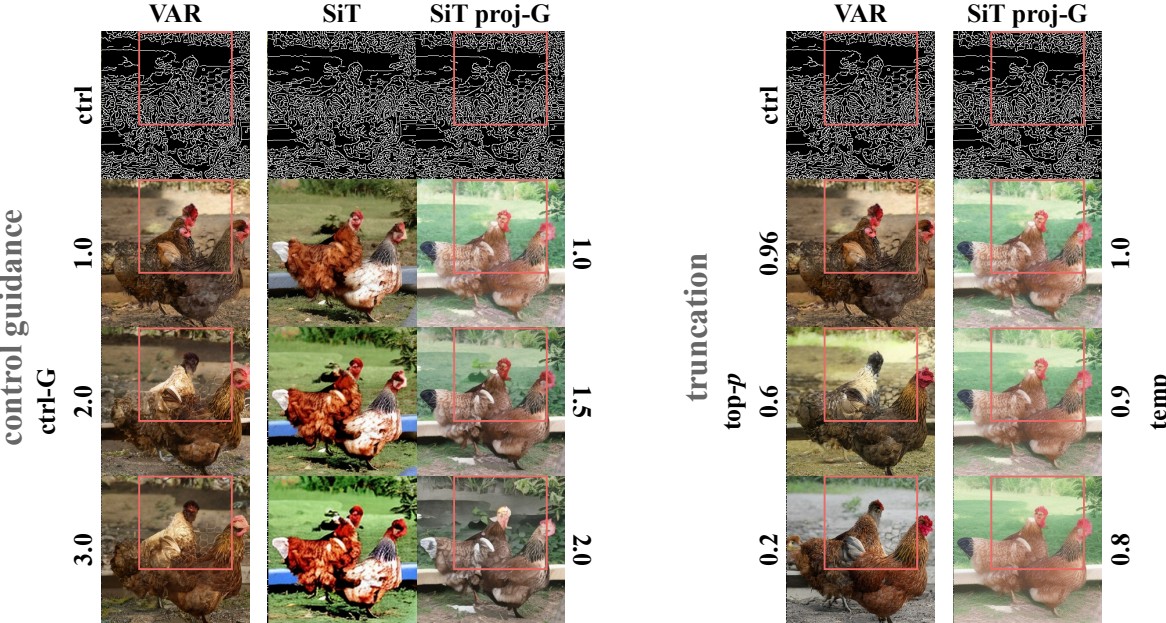

Figure 3: Visualisation of the effect of adjusting sampling parameters, CFG=3.0 (please zoom in for details). **Left (ctrl-G):** consistency to the canny map is visibly improved, but this may come at the cost of image quality, *e.g.* over-emphasised edges or incongruous shapes, *e.g.* note the treatment of what was originally a hexagonal metal wire mesh in the original control image. SiT suffers from saturation artefacts; applying projected guidance (proj-G) ameliorates this issue. **Right (distribution truncation):** Top-$p$ softmax truncation visibly improves VAR consistency without hurting generation quality. Temperature scaling the score has little visible effect on the generation.

## 5.1 Control Guidance (ctrl-G)

Consider the idea of CFG introduced in Eq. (6); it can be easily extended to the spatial control condition $c$, to concentrate sampling density in regions where $p(c|z)$ is implicitly higher, potentially improving control consistency,

$$p_{\boldsymbol{\theta}}^{\text{guid}}(\boldsymbol{z}) = \frac{1}{Z} p_{\boldsymbol{\theta}}(\boldsymbol{z}|\boldsymbol{y},\boldsymbol{c}) \underbrace{\left[\frac{p_{\boldsymbol{\theta}}(\boldsymbol{z}|\boldsymbol{y},\boldsymbol{c})}{p_{\boldsymbol{\theta}}(\boldsymbol{z}|\boldsymbol{c})}\right]^{\gamma_y-1}}_{\text{class/prompt}} \underbrace{\left[\frac{p_{\boldsymbol{\theta}}(\boldsymbol{z}|\boldsymbol{c})}{p_{\boldsymbol{\theta}}(\boldsymbol{z})}\right]^{\gamma_c-1}}_{\text{spatial control}}, \qquad p(\boldsymbol{y}|\boldsymbol{z},\boldsymbol{c}) \propto \frac{p(\boldsymbol{z}|\boldsymbol{y},\boldsymbol{c})}{p(\boldsymbol{z}|\boldsymbol{c})}, \quad p(\boldsymbol{c}|\boldsymbol{z}) \propto \frac{p(\boldsymbol{z}|\boldsymbol{c})}{p(\boldsymbol{z})}, \quad (8)$$

giving updated guidance equations for diffusion score estimates $s$ and VAR logits $v$,

$$\boldsymbol{s}_{\boldsymbol{\theta}}^{\text{guid}}(\boldsymbol{z}_t|\boldsymbol{y},\boldsymbol{c}) = \boldsymbol{s}_{\boldsymbol{\theta}}(\boldsymbol{z}_t|\boldsymbol{y},\boldsymbol{c}) \quad + (\gamma_y-1)\left[\boldsymbol{s}_{\boldsymbol{\theta}}(\boldsymbol{z}_t|\boldsymbol{y},\boldsymbol{c}) - \boldsymbol{s}_{\boldsymbol{\theta}}(\boldsymbol{z}_t|\boldsymbol{c})\right] + (\gamma_c-1)\left[\boldsymbol{s}_{\boldsymbol{\theta}}(\boldsymbol{z}_t|\boldsymbol{c}) - \boldsymbol{s}_{\boldsymbol{\theta}}(\boldsymbol{z}_t)\right] \quad (9)$$

$$\boldsymbol{v}_{\boldsymbol{\theta}}^{\text{guid}}(\boldsymbol{r}|\boldsymbol{y},\boldsymbol{c}) = \boldsymbol{v}_{\boldsymbol{\theta}}(\boldsymbol{r}|\boldsymbol{y},\boldsymbol{c}) \quad + (\gamma_y-1)\left[\boldsymbol{v}_{\boldsymbol{\theta}}(\boldsymbol{r}|\boldsymbol{y},\boldsymbol{c}) - \boldsymbol{v}_{\boldsymbol{\theta}}(\boldsymbol{r}|\boldsymbol{c})\right] + (\gamma_c-1)\left[\boldsymbol{v}_{\boldsymbol{\theta}}(\boldsymbol{r}|\boldsymbol{c}) - \boldsymbol{v}_{\boldsymbol{\theta}}(\boldsymbol{r})\right]. \quad (10)$$

In the above equations, $\gamma_y$ controls the extent of CFG, whilst $\gamma_c$ influences the control guidance (ctrl-G). Extending CFG to additional conditions has been proposed in prior work (Brooks et al., 2023; Li et al., 2024b), however, its effect on spatial control has not been well explored. For our application, prior work does not vary over $(\gamma_y, \gamma_c)$ independently for evaluation, *e.g.* Xiao et al. (2025) report a single parameter setting whilst Li et al. (2024b) set $\gamma_y = \gamma_c$. We also note that using ctrl-G incurs extra cost, requiring three model inferences rather than the two needed for CFG.

**Oversaturation for diffusion/flow-based generation.** Applying ctrl-G to SiT quickly runs into an issue – generations become highly saturated, shown in Fig. 3. Thankfully, Sadat et al. (2025) recently propose *projected* guidance (proj-G) to deal with contrast/saturation issues arising in large CFG scenarios. Applying their approach to ctrl-G, we get,

$$\boldsymbol{s}_{\boldsymbol{\theta}}^{\text{proj-guid}}(\boldsymbol{z}_t|\boldsymbol{y},\boldsymbol{c}) = [\alpha_t \boldsymbol{z}_{\boldsymbol{\theta}}^{\text{proj-guid}}(\boldsymbol{z}_t|\boldsymbol{y},\boldsymbol{c}) - \boldsymbol{z}_t]/\sigma_t^2, \quad \boldsymbol{z}_{\boldsymbol{\theta}}^{\text{proj-guid}}(\boldsymbol{z}_t|\boldsymbol{y},\boldsymbol{c}) = \boldsymbol{z}_{\boldsymbol{\theta}}(\boldsymbol{z}_t|\boldsymbol{y},\boldsymbol{c}) + [\boldsymbol{g} - (\boldsymbol{g} \cdot \hat{\boldsymbol{z}})\hat{\boldsymbol{z}}], \quad (11)$$

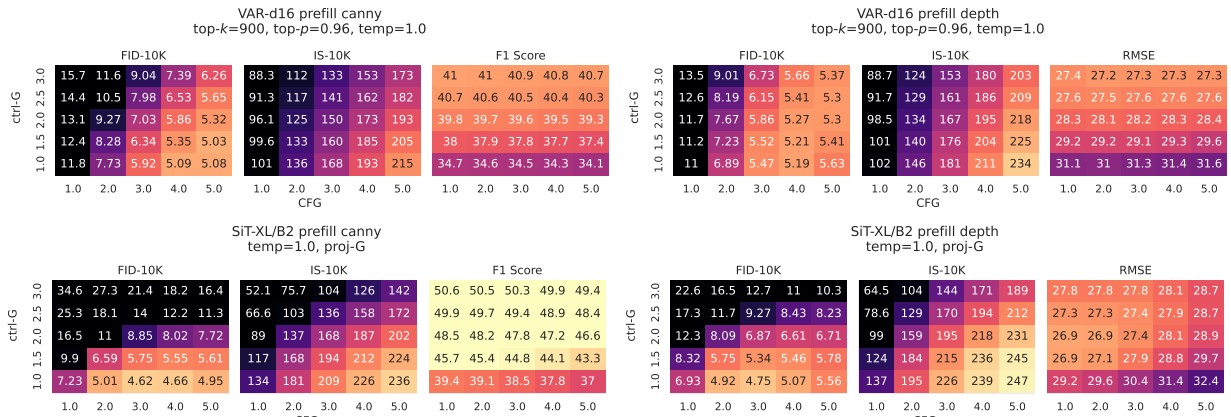

Figure 4: The effect of CFG and Ctrl-G on conditional generation. Brighter means better. CFG generally improves generation quality according to FID↓ and IS↑, with a slight decrease in control consistency (F1↑, RMSE↓). **Ctrl-G significantly improves consistency, but introduces a trade-off against generation quality.**

$$\text{where} \quad \boldsymbol{g} = (\gamma_y - 1)\left[\boldsymbol{z_\theta}(\boldsymbol{z}_t|\boldsymbol{y}, \boldsymbol{c}) - \boldsymbol{z_\theta}(\boldsymbol{z}_t|\boldsymbol{c})\right] + (\gamma_c - 1)\left[\boldsymbol{z_\theta}(\boldsymbol{z}_t|\boldsymbol{c}) - \boldsymbol{z_\theta}(\boldsymbol{z}_t)\right], \qquad \hat{\boldsymbol{z}} = \frac{\boldsymbol{z_\theta}(\boldsymbol{z}_t|\boldsymbol{y}, \boldsymbol{c})}{||\boldsymbol{z_\theta}(\boldsymbol{z}_t|\boldsymbol{y}, \boldsymbol{c})||} \quad (12)$$

The model is reparameterised to *data* ($\boldsymbol{z}_0$) prediction and only the component of guidance that is *orthogonal* to the fully conditioned data estimate $\boldsymbol{z_\theta}(\boldsymbol{z}_t|\boldsymbol{y}, \boldsymbol{c})$ is used. Sadat et al. (2025) show that this orthogonal component is chiefly responsible for quality improvements, whilst the other (parallel) component increases saturation, as it intuitively extends the data prediction $\boldsymbol{z_\theta}$. We find this successfully fixes the saturation issue, without otherwise negatively affecting generations, hence we adopt proj-G for all our following SiT experiments. We also note that the *constrained* distribution over the discretised latent space of a VQVAE modelled by VAR is not susceptible to these saturation issues.

> **Takeaway:** For flow/diffusion models, control guidance (Eq. (9)) may lead to oversaturation. Applying projected guidance (Sadat et al., 2025) (*i.e.* only using the orthogonal component Eq. (11)) assuages this issue.

**Results.** Fig. 4 shows how image generation quality and control consistency vary with both CFG ($\gamma_y$) and ctrl-G ($\gamma_c$). Compared to the default settings used in Fig. 2, we find that increasing CFG generally improves both IS and FID (although too high a scale may hurt FID), but slightly decreases consistency. Interestingly, in the case of VAR, higher CFG *improves* FID over the no-control model ($\sim 5.7$ to $\sim 5.1$), demonstrating that providing a weaker model with spatial cues can boost generation quality. On the other hand, ctrl-G is able to drastically improve consistency, but this is traded off against worse generation quality. This is visualised in the left of Fig. 3, where we see that the canny edge map is better conformed to, but at the cost of over-emphasised edges and incongruous generation for larger ctrl-G. Intuitively, ctrl-G only targets the probability of the *control* given the generation $p(\boldsymbol{c}|\boldsymbol{z})$ (Eq. (8)), which may not align with image *quality*. This trade-off aligns with the results reported for image *editing* in (Brooks et al., 2023). We note that the F1 values of $\sim 50$ for SiT are *significantly higher* than the values commonly found in the literature (Li et al., 2025; Xiao et al., 2025), demonstrating the large effect that this *generation-time* adjustment has on consistency. Additionally, ctrl-G incurs the extra inference cost of needing to run the no-control model $p_{\boldsymbol{\theta}}(\boldsymbol{z})$, reducing latency and throughput:[6]

|  | VAR-d16 prefill + CFG | VAR-d16 prefill + CFG + ctrl-G | SiT-XL/2 prefill + CFG | SiT-XL/2 prefill + CFG + ctrl-G |
|---|---|---|---|---|
| #params | 310M | 310M (+ 310M) | 675M | 675M (+ 675M) |
| lat↓ (s) bs=1 | 0.31 | 0.55 | 3.4 | 5.9 |
| TP↑ (img/s) bs=16 | 12 | 8.4 | 0.35 | 0.24 |

> **Takeaway:** Control guidance greatly improves control consistency, but there is a trade-off against image quality and inference cost. CFG has a small negative effect on consistency but generally improves generation quality.

---

[6]We use the pretrained ImageNet model for $p_{\boldsymbol{\theta}}(\boldsymbol{z})$ *in series* with the control-prefill model. Better latency and parameter efficiency could be achieved by training the control-prefill model with empty control $\boldsymbol{c} = \varnothing$ and parallelising like CFG, although the FLOPs increase would still remain.

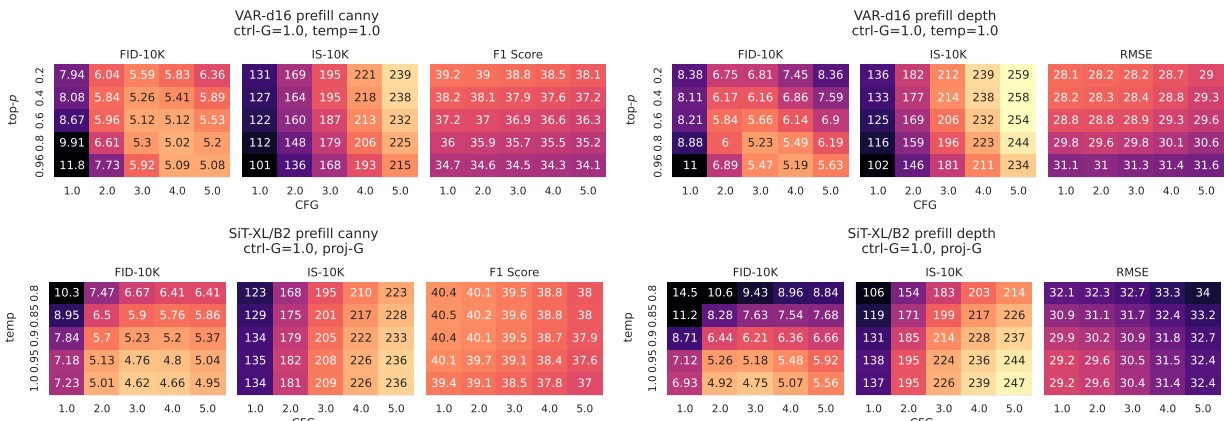

Figure 5: Effect of CFG and distribution truncation on conditional generation. Brighter means better. Again, CFG generally improves generation quality, with a slight decrease in consistency. **Top-$p$ softmax truncation improves both generation quality and consistency for VAR**, although aggressive truncation may reduce diversity and hurt FID. Score temperature scaling does not produce any meaningful benefit for SiT.

Finally we note that in ControlVAR (Li et al., 2024b), when performing guidance (with class and control conditioning), all guidance scales $\gamma$ are set to be equal. However, Fig. 4 shows that this setting may lead to excessive degradation in generation quality, demonstrating the importance of *separately tuning* $\gamma_y$ (CFG) and $\gamma_c$ (ctrl-G).

## 5.2 Sampling Distribution Truncation

Another intuitive, but unexplored way of potentially improving control consistency, is to directly truncate/concentrate the sampling distribution. This intuitively assumes that the high density/probablity regions predicted by the model $p_\theta(\boldsymbol{z}|\boldsymbol{y}, \boldsymbol{c})$ will most conform to the input spatial control. We select one method for each of VAR and SiT.

**Top-$p$ sampling for VAR.** As VAR samples from a *softmax* output $\boldsymbol{\pi}$, there are a number of options to concentrate probability mass (temperature, top-$p$, top-$k$). In this work, we focus on top-$p$ sampling (Holtzman et al., 2020), which has been shown to effectively improve the coherence of language generation. It directly truncates the softmax distribution $\boldsymbol{\pi}$,

$$\pi_k^{\text{top-}p} = \frac{\pi_k}{\sum_{i \in V_p} \pi_i} \text{ if } k \in V_p, \text{ otherwise } 0 \qquad \text{where } V_p = \text{the smallest set of indices } V \text{ s.t. } \sum_{i \in V} \pi_i \geq p, \quad (13)$$

effectively discarding the tail $1 - p$ of the softmax. Intuitively, one would expect the most probable vector codes, predicted by the model's softmax at each token location, to best conform to the spatial control condition.

**Score temperature scaling for diffusion/flow models.** In the case of diffusion, temperature scaling the score (Dhariwal & Nichol, 2021; Ingraham et al., 2023; Karras et al., 2024), can be intuited as raising the marginal distributions to power $1/\tau$, also focusing sampling on higher density regions,

$$\frac{1}{\tau_t} \nabla_{\boldsymbol{z}_t} \log \ p(\boldsymbol{z}_t) = \nabla_{\boldsymbol{z}_t} \log \left[ p(\boldsymbol{z}_t)^{\frac{1}{\tau_t}} \right], \quad \text{where} \quad 1/\tau_t = \frac{(\alpha_t^2 + \sigma_t^2)\frac{1}{\tau_0}}{\alpha^2 + \frac{\sigma^2}{\tau_0}}. \quad (14)$$

We adopt the annealing schedule from Ingraham et al. (2023) for $1/\tau_t$ that samples from the temperature-scaled data distribution *assuming it is Gaussian*, where $\tau_0$ is the temperature at $t = 0$ (clean data), labelled "temp" in our experiments. This choice is motivated by the poor generation results from Dhariwal & Nichol (2021) for constant $\tau$.

**Results.** Figs. 3, 5 and 8 show how top-$p$ and temperature scaling affect generation quality and control consistency alongside CFG. We see that CFG behaves similarly to Fig. 4, generally improving generation quality and slightly hurting consistency. Notably, truncation via top-$p$ is able to improve *both* generation quality and control consistency (although excessive truncation seems to hurt FID likely due to reduced diversity). Fig. 8 (Appendix A) further shows that the consistency improvements can be *stacked* with ctrl-G. This demonstrates that top-$p$ is an important parameter to adjust for softmax-based generative models, providing improvements in all fronts *without any additional cost*. In

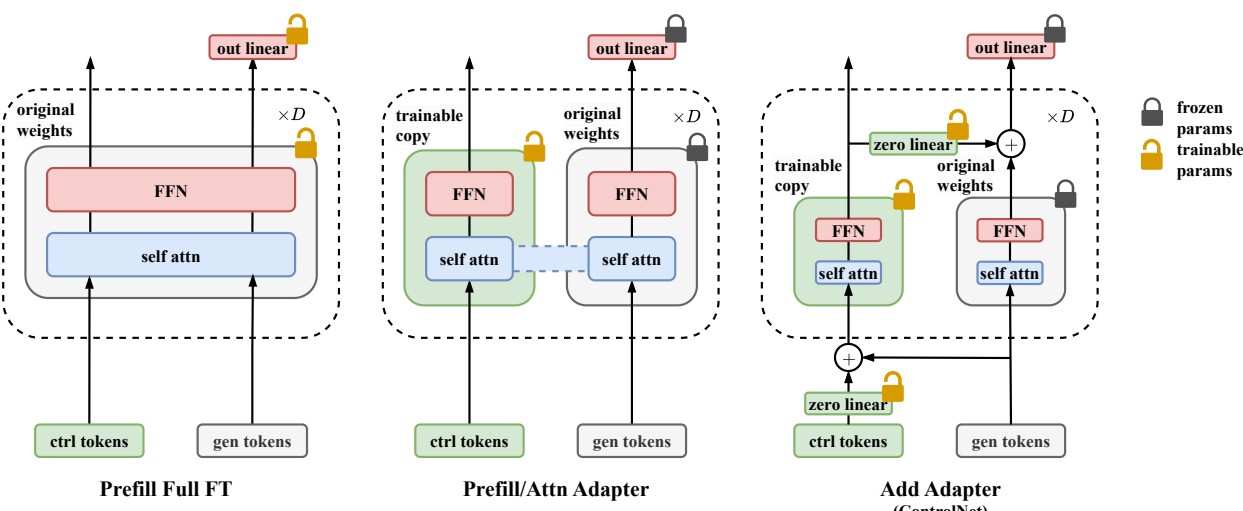

Figure 6: Illustration of adapter approaches considered in this work. Compared to full finetuning with prefill (left), adapters (centre, right) freeze the original weights, using a trainable copy to process control tokens and communicate spatial conditioning to generation tokens at each layer. The prefill/attention adapter communicates flexibly via sharing key-value pairs with the generation tokens, whilst the addition adapter directly injects information via zero-initialised linear layers (Zhang et al., 2023), requiring each generation token to be spatially matched to a control token.

contrast, temperature scaling is unable to meaningfully improve consistency, leading to a small increase for canny, and no improvement for depth, whilst simultaneously hurting generation quality. Compared to the results for ctrl-G, this suggests that naively scaling the score (without changing its direction) is insufficient to improve consistency. One possible intuition, is that score temperature scaling occurs *uniformly* over the spatial dimensions, *i.e.* it is a global transformation, and so it may have limited *spatial/local* influence, whilst ctrl-G varies in space. On the other hand, top-$p$ explicitly truncates at each token *location*, allowing it to influence spatial consistency.

> **Takeaway:** Distributional truncation of the softmax (top-$p$ sampling), can improve both generation quality and control consistency for VAR at inference time for no extra cost. Conversely, truncation via temperature scaling of the score does not meaningfully benefit the consistency or generation quality of diffusion/flow models.

We finally note that Figs. 4, 5 and 8 demonstrate that *meaningfully improved performance* can be easily obtained by the sampling adjustments explore in this section (compared to the default settings used in Fig. 2).

> **Takeaway:** Classifier-free guidance, control guidance and softmax truncation offer meaningful additional "knobs" that practitioners can adjust to improve the characteristics of their generations *at sampling time*.

## 6 Adapters for Spatial Control

In the previous sections, we did not consider adapter-style approaches, typified by ControlNet (Zhang et al., 2023), where a pretrained image generation model's weights are *frozen* and an additional module is trained for spatial control. In this section, we aim to clarify and empirically demonstrate to readers the motivation for adapters, positioning them relative to approaches where all parameters are trained. Adapter-based approaches are motivated by the problem setting of *adding* spatial control to existing powerful pre-trained foundation models such as Stable Diffusion (Rombach et al., 2022; Podell et al., 2024; Esser et al., 2024) and Flux (Labs, 2024) whilst mitigating the potential that fine-tuning on limited/undiverse downstream control would cause degradation in general generation ability and "forgetting" of concepts in the pretraining data (Zhang et al., 2023). However, we highlight that this problem setting may not apply to all practitioners. For example, an adapter-based approach may not actually be the most suitable if the aim is to endow a single generative foundation model with many abilities from the ground up (Chen et al., 2025b; Xiao et al., 2025; Zhang et al., 2025b), or if fine-tuning data is rich and diverse enough that forgetting is not a concern. We note that this nuance may become lost in the literature, *e.g.* Li et al. (2025) benchmark their approach, where all parameters are trained, against adapters without discussing

**VAR-d16 (left)**

| model (sampling params) | method | ctrl | #params base+adapter | FID↓ 10K | IS↑ 10K | F1↑ | RMSE↓ | lat↓ (s) bs=1 | TP↑ (img/s) bs=16 |
|---|---|---|---|---|---|---|---|---|---|
| ImageNet-val (ctrl source) | | | | 4.44 | 187 | | | | |
| VAR-d16 CFG=2.5, temp=1, top-p=0.96, top-k=900 | none | | 310M | 5.71 | 214 | | | 0.26 | 17 |
| VAR-d16 CFG=3.0, temp=1, top-p=0.6, top-k=900 — ctrl-G =1.0 | prefill | canny depth | 310M | 5.12 / 5.66 | 187 / 207 | 36.9 | 28.9 | 0.31 | 12 |
| | add adapter (CtrlNet) | canny depth | 310M + 220M | 5.51 / 6.08 | 192 / 217 | 33.6 | 31.1 | 0.55 | 7.9 |
| | prefill adapter | canny depth | 310M + 202M | 4.98 / 5.64 | 201 / 208 | 33.0 | 29.7 | 0.31 | 12 |
| ctrl-G =1.5 | prefill | canny depth | 310M×2 | 5.40 / 5.59 | 180 / 199 | 39.9 | 27.4 | 0.55 | 8.4 |
| | add adapter (CtrlNet) | canny depth | 310M + 220M | 6.10 / 5.86 | 174 / 207 | 37.3 | 29.3 | 0.80 | 6.3 |
| | prefill adapter | canny depth | 310M + 202M | 5.15 / 5.47 | 190 / 201 | 35.9 | 27.9 | 0.56 | 8.5 |

**SiT-XL/2 (right)**

| model (sampling params) | method | ctrl | #params base+adapter | FID↓ 10K | IS↑ 10K | F1↑ | RMSE↓ | lat↓ (s) bs=1 | TP↑ (img/s) bs=16 |
|---|---|---|---|---|---|---|---|---|---|
| ImageNet-val (ctrl source) | | | | 4.44 | 187 | | | | |
| SiT-XL/2 CFG=1.5, Euler ODE, steps=64 | none | | | 4.47 | 228 | | | 3.0 | 0.39 |
| SiT-XL/2 CFG=3.0, Euler-ODE, steps=64, proj-G — ctrl-G =1.0 | prefill | canny depth | 675M | 4.62 / 4.75 | 209 / 226 | 38.5 | 30.4 | 3.4 | 0.35 |
| | add adapter (CtrlNet) | canny depth | 675M + 485M | 5.02 / 4.84 | 226 / 241 | 33.3 | 33.5 | 6.2 | 0.19 |
| | prefill adapter | canny depth | 675M + 446M | 4.52 / 4.62 | 235 / 251 | 30.8 | 32.7 | 3.4 | 0.35 |
| ctrl-G =1.5 | prefill | canny depth | 675M×2 | 5.75 / 5.34 | 194 / 215 | 44.8 | 27.9 | 5.9 | 0.24 |
| | add adapter (CtrlNet) | canny depth | 675M + 485M | 6.00 / 5.51 | 210 / 228 | 40.5 | 30.5 | 8.1 | 0.15 |
| | prefill adapter | canny depth | 675M + 446M | 5.39 / 5.17 | 218 / 227 | 37.1 | 29.7 | 5.8 | 0.24 |

Table 1: Performance of adapter-based approaches compared to the prefill baseline. **(a)** Adapters have worse control consistency (F1,RMSE) compared to prefill+finetune, although generation quality is comparable. **(b)** The addition adapter is better for canny conditioning, whilst the prefill adapter is better for depth conditioning. **(c)** Adjusting sampling parameters (such as ctrl-G) can make up some of the difference in control consistency. **(d)** The prefill adapter is a little more parameter efficient (no zero layers) and considerably more inference efficient than the addition adapter. **(e)** Adapters enjoy no parameter overhead for ctrl-G, as the control-free base model weights are frozen and available.

the above. The following experiments aim to highlight the differences between training all parameters versus using an adapter, so that practitioners can have a clearer view of which style of approach is more suitable for their use-case.

## 6.1 Adapter Architectures

We consider two different adapter architectures, that supply spatial control information to the base model via two different mechanisms. We note that in both cases, we train copies of the base model's transformer blocks, only fine-tuning the FFN, self-attention modules and learned positional embeddings, leaving adaptive layer norm frozen.

**Add adapter.** Illustrated in the right of Fig. 6, this approach adapts the ControlNet (Zhang et al., 2023) architecture to transformers, like in Chen et al. (2024a), training a copy of the base model to additively inject spatial conditioning information at each layer, via zero-initialised linear layers, whilst the original weights are performing generation. Inference for each generation token is *matched* by inference on a corresponding control token.

**Attention/prefill adapter.** Illustrated in the centre of Fig. 6, this approach retains the prefill paradigm of conditioning from the baseline, and trains a separate copy of the transformer blocks to be used during condition prefilling, leaving generation to the original model weights. Although the adapter and base model have different attention weight matrices, generation tokens are able to attend back to the keys and values produced by the control adapter.

These two approaches aim to broadly represent two distinct potential options for adapter-based approaches to spatial conditioning, i.e., attention and additive injection. Our experiments are of course not exhaustive, and there is an exponentially large space of specific design choices that could be explored: selective freezing, the use of low-rank adaptation (LoRA) (Hu et al., 2022), the depth of the adapter, the location and direction of the exchange of conditioning information *etc*. However, we hope that our experiments can still provide practical high-level research takeaways to the community.

## 6.2 Results

Based on the exploration of the previous section, we select a set of sampling parameters that aim to balance generation quality with control consistency for our remaining experiments. They are specified in Tab. 1 and Fig. 7. Additional generation examples for the following experiments can be found in Appendix B.

**Adapters vs prefill baseline (training on full data).** Tab. 1 shows results comparing the performance of the two adapter approaches to the prefill + finetuning baseline. We find that generally speaking adapters have *worse* control

consistency, but have comparable generation quality. Intuitively, an adapter is less *flexible* to influence the generation process, limiting its ability to closely follow conditioning, but also retaining the original model's generation capabilities. This aligns with existing research on LoRA in language modelling (Biderman et al., 2024). The addition adapter has the advantage for canny conditioning, whilst the prefill adapter is superior for depth conditioning, although both are worse than full finetuning. We also demonstrate that adjusting the sampling parameters of an adapter (in this case ctrl-G) can make up some of difference in control consistency, although the associated tradeoffs discussed in Sec. 5 still exist. We note that this suggests the comparisons made in ControlAR (Li et al., 2025) and Omnigen (Xiao et al., 2025) may not be entirely fair, as their approaches train all parameters, but they compare against adapter-based approaches.

> **Takeaway:** Adapters, where the generative model is frozen, can learn spatial conditioning. Control consistency is consistently worse than prefill + finetuning all parameters, although good generation quality is maintained.

We find that the prefill adapter is slightly more parameter efficient (due to not needing any additional zero-initialised layers), although it is of course possible to explore other ways of improving parameter efficiency, such as reducing the depth/number of blocks in the adapter or using LoRA (Hu et al., 2022).[7] We note that prefilling (adapter or not) is considerably more inference efficient compared to the addition adapter based on ControlNet (Zhang et al., 2023). The former is able to utilise a KV-cache, meaning that inference over control tokens only happens once per generation, whilst in the latter case, each generation token forward pass is paired with a spatially aligned control token forward pass, significantly increasing inference costs. This difference is especially apparent for SiT, where the KV-cache can be re-used for all (64) denoising steps. We note that Tan et al. (2025a;b) make a similar observation when designing their method for conditional generation built around FLUX (Labs, 2024).

> **Takeaway:** Prefilling (adapter or not) with control tokens is considerably more inference efficient than adapters that are run each generation step (*e.g.* ControlNet (Zhang et al., 2023)) due to the re-use of the control KV-cache.

We also note that since adapters keep the original (no control) model's weights intact, there is no *parameter* overhead for ctrl-G. As such it can be readily used and considered for many existing adapter-based systems. In Appendix C we demonstrate this, showing that applying ctrl-G to ControlNet++ (Li et al., 2024a) out of the box results in similar qualitative and quantitative trade offs to Sec. 5.1 without any need for additional training (Fig. 14).

> **Takeaway:** Ctrl-G can be used out-of-the-box with no parameter overhead for adapter-based approaches, meaning it can currently be readily applied to existing models/adapters to adjust generation behaviour.

**Forgetting.** In order to investigate the phenomenon of "forgetting" we finetune and train adapters on a semantically pruned version of ImageNet where all samples with labels that are children of the WordNet (Miller et al., 1990) node "organism" are removed. We train for the same number of iterations as previously ($\sim$ 50K). Results are shown in Figs. 7 and 9. We find that training all VAR parameters exhibits a collapse in generation quality, where the model "forgets" how to generate "organism" classes outside of the control finetuning data. Conversely, we observe that adapters are able to mitigate this effect, successfully generalising control-conditioned generation to classes unseen in the control data (although present in the base model's pretraining data). SiT interestingly is able to maintain its generation quality compared to finetuning on the full dataset. We hypothesise this robustness may be due to the larger size (675M vs 310M parameters) and longer pretraining (800 vs 200) of the SiT model (Yu et al., 2025) compared to the VAR model (Tian et al., 2024).

> **Takeaway:** Full finetuning on insufficient control data may lead to catastrophic forgetting. Adapters are more robust to this, generalising control to parts of the data distribution seen in pre-training but not in control finetuning.

Finally, to summarise the position and practical applicability of adapter-based approaches:

> **Takeaway:** Practitioners should consider the suitability of adapters for their use case. Adapters endow *pretrained* generative models with spatial controllability, mitigate "forgetting" when fine-tuning on limited data, and are modularised as an additional set of weights. However, control consistency is *worse* than when training all parameters.

---

[7]Appendix A.2 contains a supplementary experiment that shows that LoRA can be used to improve parameter efficiency at the cost of performance.

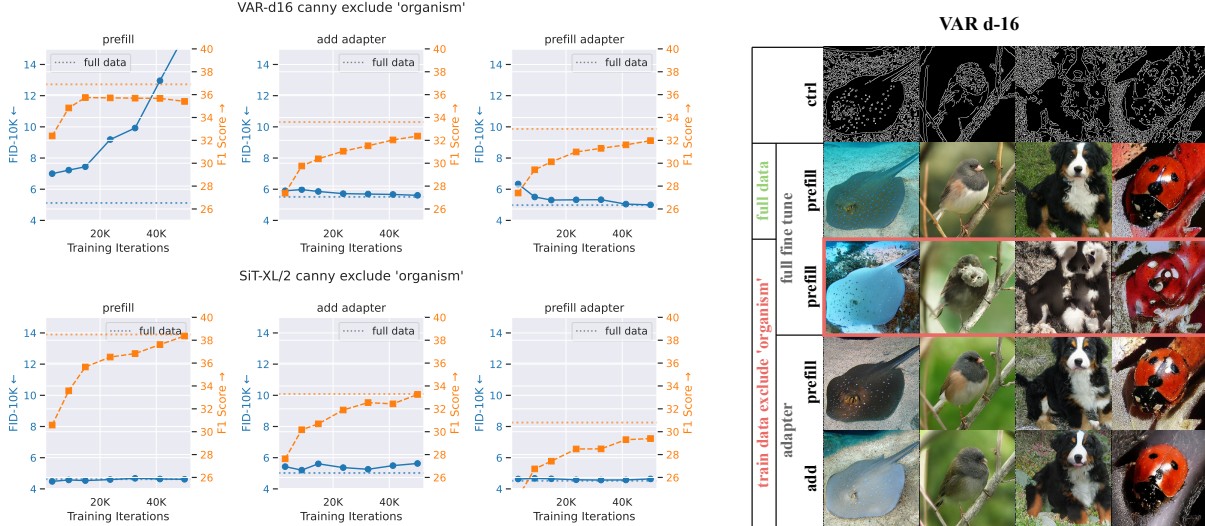

Figure 7: **Left:** Generation quality (FID↓) and control consistency (F1↑) of Canny-conditioned VAR trained on control data that excludes classes that are "organisms" according to WordNet, evaluated for the *whole* ImageNet distribution. Fully finetuning VAR (prefill) leads to a collapse in generation quality, whilst using adapters mitigates this. On the other hand, the larger SiT model manages to maintain generation quality over the whole distribution. See Fig. 9 in Appendix A for depth results. **Right:** Visualisation of generations – full finetuning exhibits catastrophic forgetting, whilst adapters are able to generalise control-conditioned generation to "organism" classes.

# 7 Concluding Remarks

In this work, we investigate spatially controlled image generation with transformers, performing controlled experiments over diffusion/flow and autoregressive models. We demonstrate that control token prefilling is a simple and well-performing baseline that can be generally applied across generation paradigms. We then find that simple sampling enhancements, such as control guidance and softmax truncation can meaningfully improve generation-control consistency. Finally, we reaffirm the practical motivation for adapter-based approaches, demonstrating that they can add spatial control to pretrained generative models, whilst maintaining generation quality/mitigating "forgetting" when finetuned on limited control data. However, this comes at the cost of inferior generation-control consistency compared to training all parameters. We hope the takeaways in this work can be useful to practitioners and researchers in the future.

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

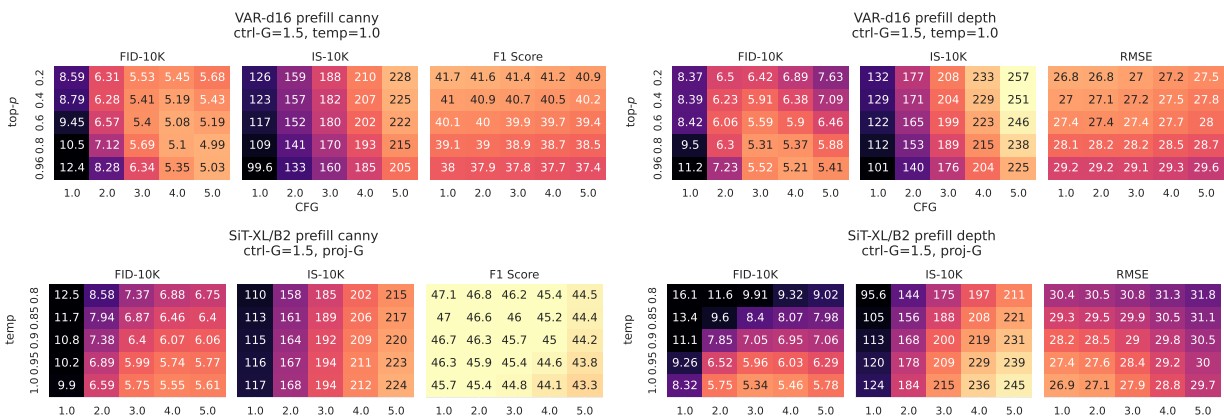

Figure 8: Effect of CFG and distribution truncation on conditional generation when ctrl-G=1.5. Brighter means better. The improvements in control consistency from softmax truncation via top-$p$ sampling stack with ctrl-G.

| model | method | ctrl | #params | metrics | | |
|---|---|---|---|---|---|---|
| (sampling params) | | | base + adapter | FID-10K↓ | IS-10K↑ | F1↑ |
| VAR-d16 | prefill | canny | 310M | 5.12 | 187 | 36.9 |
| CFG=3.0, temp=1, | prefill + LoRA (r=64) | canny | 310M + 10M | 5.57 | 185 | 33.2 |
| top-$p$=0.6, top-$k$=900, | prefill adapter | canny | 310M + 202M | 4.98 | 201 | 33.0 |
| ctrl-G=1.0 | prefill adapter + LoRA (r=64) | canny | 310M + 10M | 5.59 | 182 | 28.9 |

Table 2: Experiments with Low-Rank Adaptation (LoRA). We use rsLoRA (Kalajdzievski, 2023) and set $r = \alpha = 64$. We attach LoRA adapters to the $QKV$ matrices in each attention layer as well as the weight matrices in the feed-forward network. For prefill + LoRA the trainable weights see all tokens, whilst for prefill adapter + LoRA the trainable weights only see control tokens (Fig. 6 where the trainable parameters are replaced with LoRA). LoRA enables much lower parameter overhead, but the reduced model flexibility comes at the cost of generation quality and control consistency.

# A  Additional Experiments

## A.1  Distribution Truncation

Fig. 8 shows how distribution truncation and CFG interact when ctrl-G=1.5. It tells a similar story to Fig. 4 and also illustrates how both distribution truncation and control guidance can be adjusted together at inference time to improve control consistency. For VAR, the control consistency improvements from ctrl-G and top-$p$ stack together.

## A.2  Low Rank Adaptation (LoRA)

Tab. 2 shows an additional ablation, replacing the trainable parameters in Fig. 6 with LoRA (Hu et al., 2022) adapters. We find that LoRA can considerably improve the parameter efficiency of learning a spatial control; however, it comes at the cost of generation quality as well as control consistency. This aligns with the behaviour reported in Li et al. (2025).

## A.3  Learning with Limited Data

Fig. 9 mirrors the left of Fig. 7 but for depth conditioning. We note that excluding "organism" classes from the finetuning data consistently hurts depth consistency (RMSE↓). We hypothesise that this may be due to distributional shift between the depth maps of organism classes present at evaluation and non-organism classes used in training.

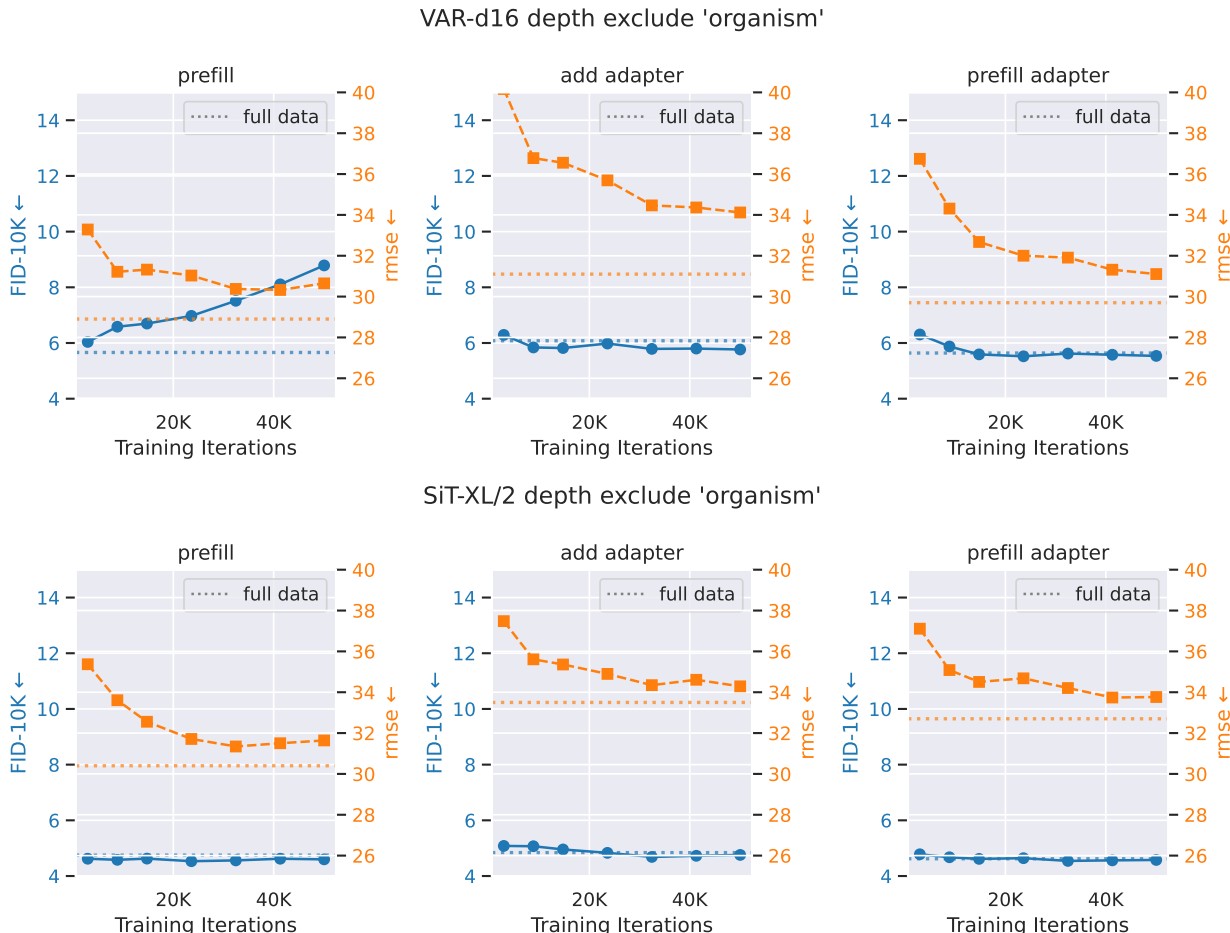

Figure 9: Generation quality (FID↓) and control consistency (RMSE↓), evaluated for the whole ImageNet distribution. Fully finetuning VAR (prefill) leads to a collapse in generation quality, whilst using adapters mitigates this. On the other hand the larger SiT model manages to maintain generation quality over the whole distribution.

## B Example Generations

We include additional example generations over various experimental settings in Figs. 10 to 13. Readers can observe the effect of ctrl-G (canny: facial features of the dog, depth: seeds/stones next to the bird) across Figs. 10 and 12. Fig. 11 show further examples of VAR "forgetting" "organism" concepts from ImageNet-pretraining. We note these are more subtle for depth (plumage on bird, stripes rather than spots on salamander).

## C Ctrl-G on Pre-trained Adapters for T2I Models

We demonstrate that ctrl-G can be directly applied to existing opensource adapters without the need for any additional training. We show the effect of increasing ctrl-G on ControlNet++ (Li et al., 2024a), a control adapter for StableDiffusion 1.5 (Rombach et al., 2022) that produces 512×512 resolution text-prompt-conditioned images. Fig. 14 illustrates qualitative improvements in control consistency (the shape of vegetation on the building, the curve of the window), whilst also quantitatively showing the control consistency vs FID tradeoff that ctrl-G introduces, aligning with earlier results in Fig. 4. We use the evaluation code of Li et al. (2024a)[8] and calculate FID between the validation and generation images.

---

[8] https://github.com/liming-ai/ControlNet_Plus_Plus

prefill, ctrl-G=1.0

prefill, ctrl-G=1.5

add adapter, ctrl-G=1.0

add adapter, ctrl-G=1.5

prefill adapter, ctrl-G=1.0

prefill adapter, ctrl-G=1.5

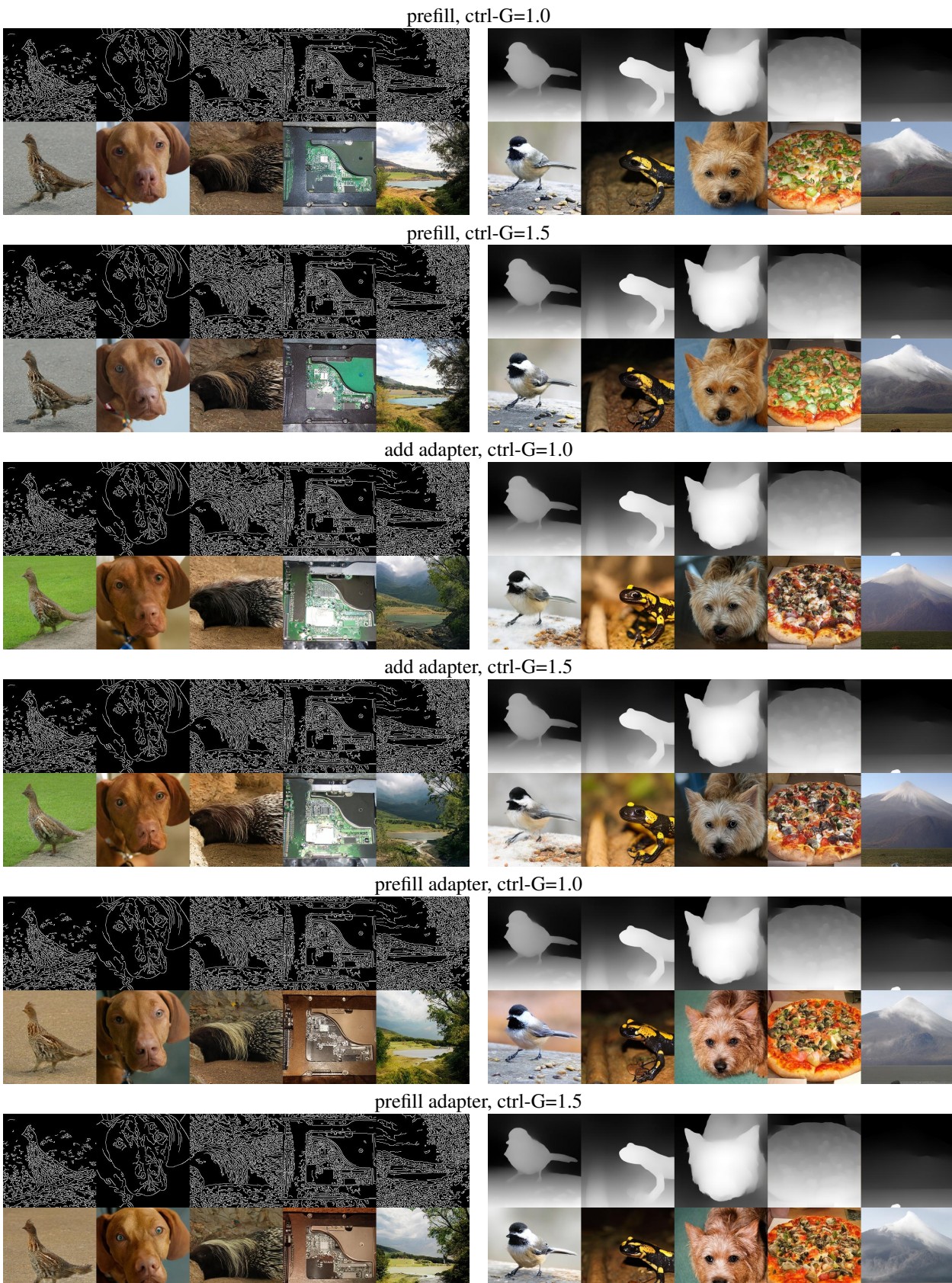

Figure 10: Additional examples of VAR generations.
(CFG=3.0, temp=1, top-$p$=0.6, top-$k$=900)

prefill

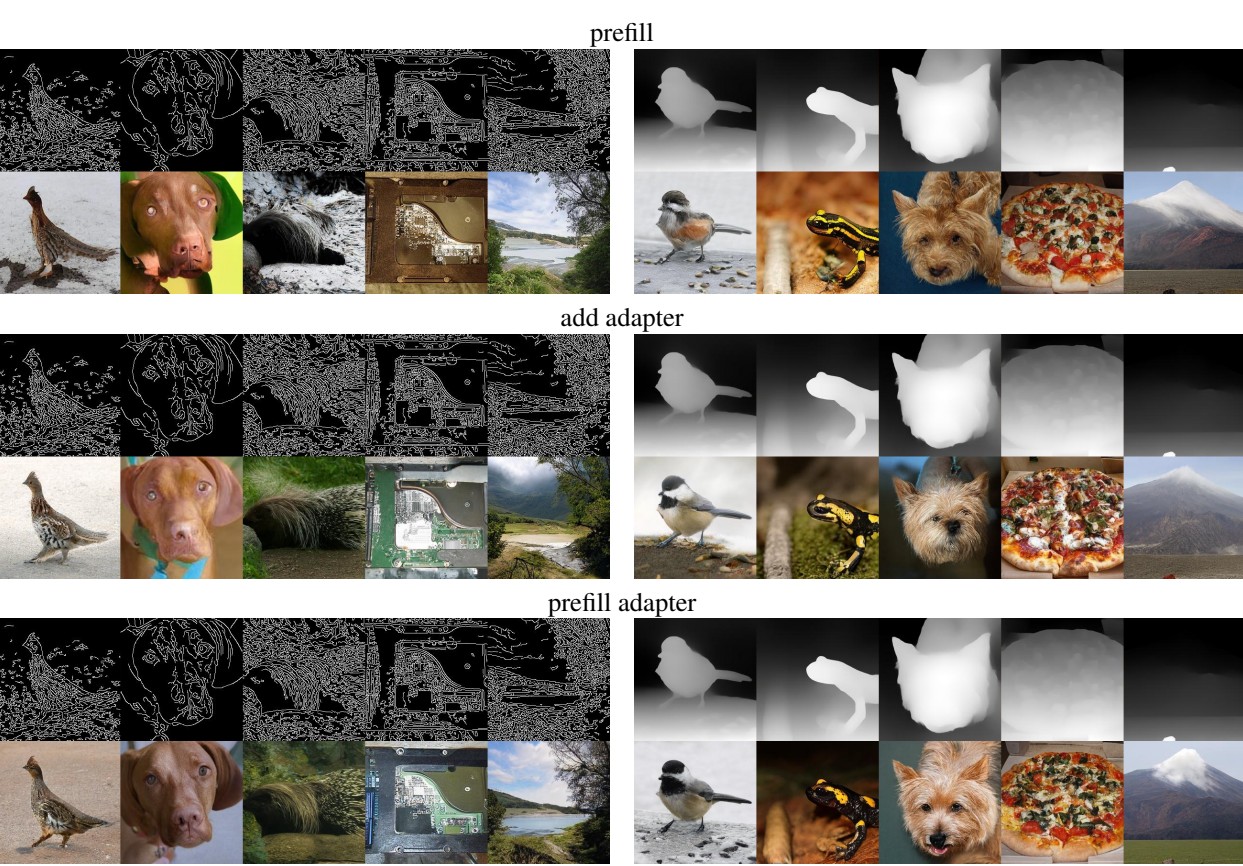

add adapter

prefill adapter

Figure 11: Additional examples of VAR generations with limited control data – "organism" classes excluded. (CFG=3.0, temp=1, top-$p$=0.6, top-$k$=900, ctrl-G=1.0)

prefill, ctrl-G=1.0

prefill, ctrl-G=1.5

add adapter, ctrl-G=1.0

add adapter, ctrl-G=1.5

prefill adapter, ctrl-G=1.0

prefill adapter, ctrl-G=1.0

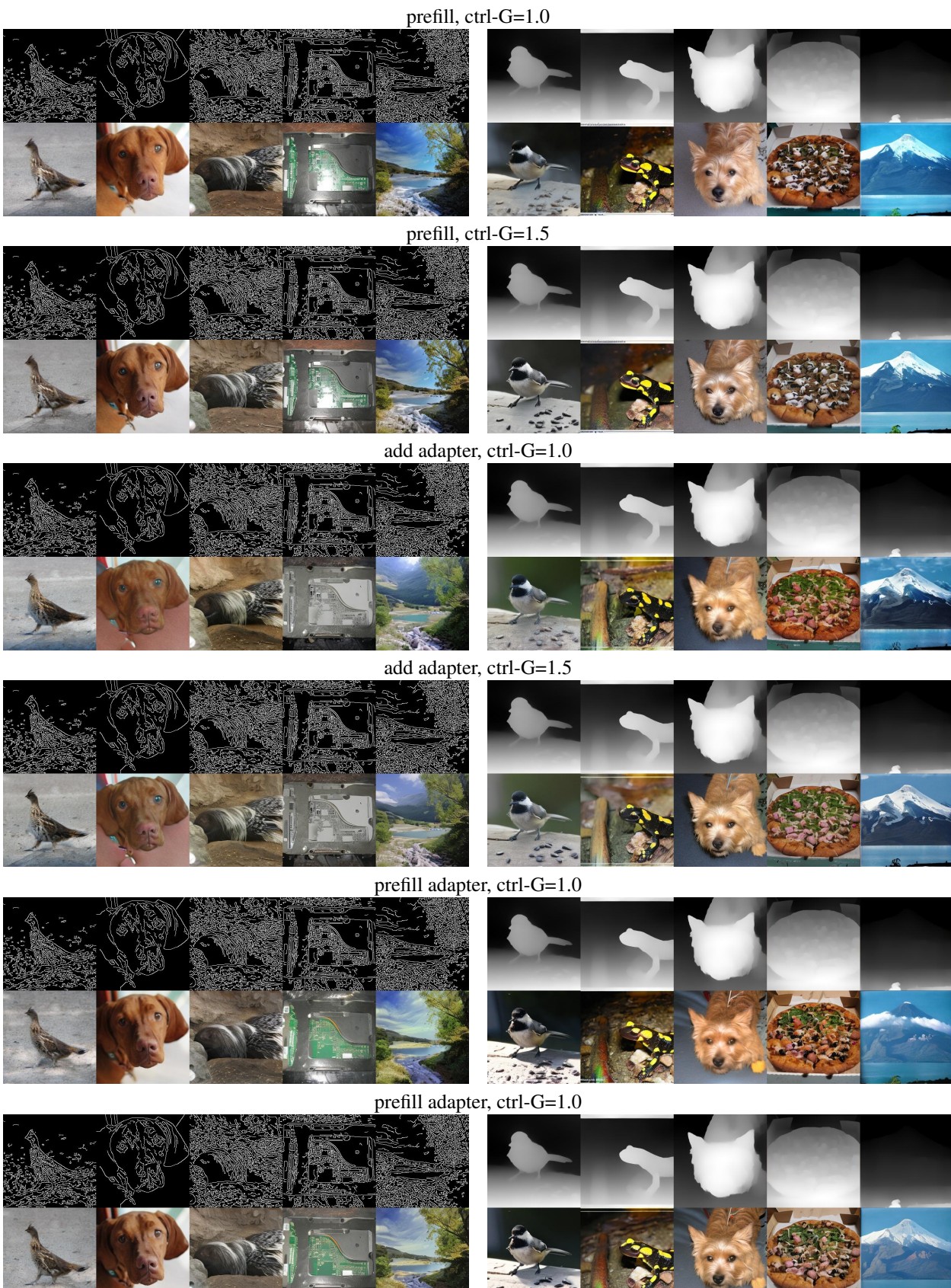

Figure 12: Additional examples of SiT generations. (CFG=3.0, temp=1.0, Euler ODE, steps=64, proj-G)

prefill

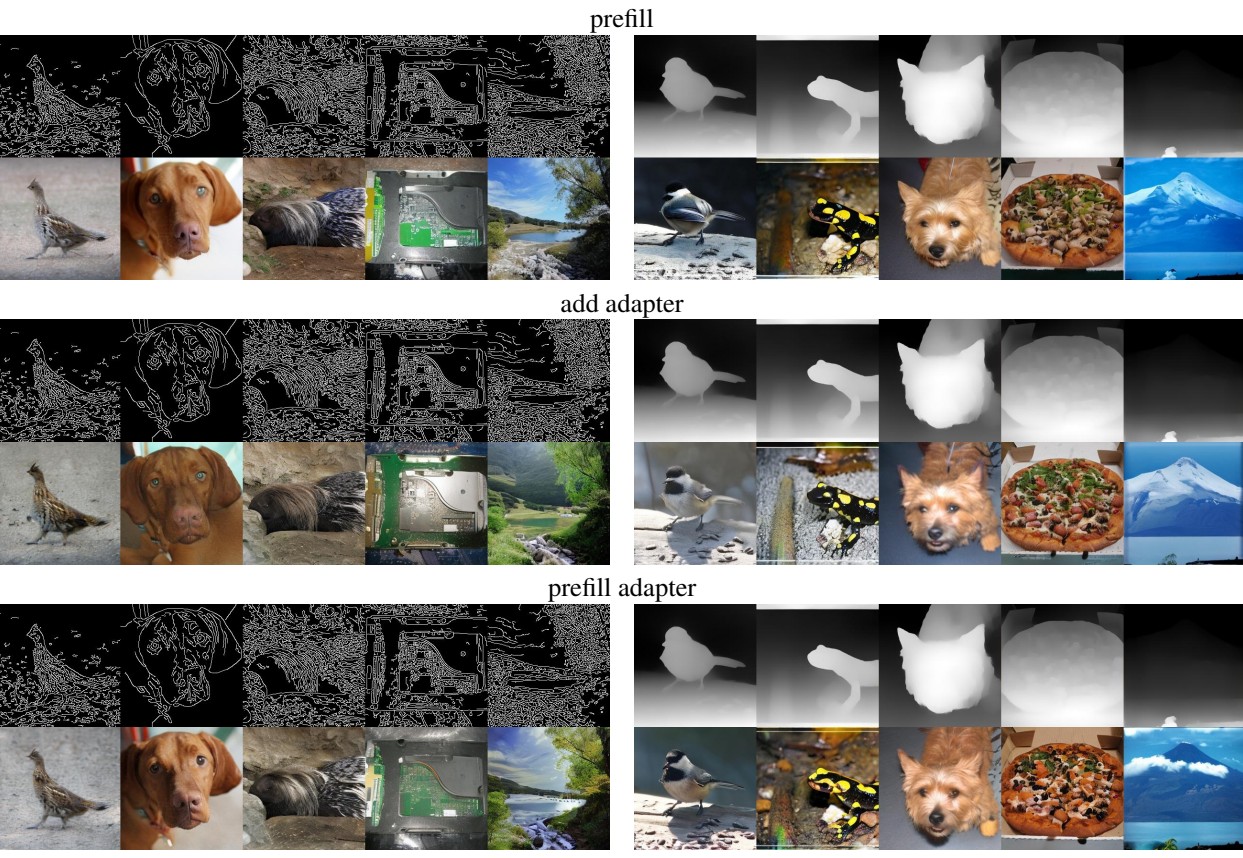

Figure 13: Additional examples of SiT generations with limited control data – "organism" classes excluded. (CFG=3.0, temp=1.0, Euler ODE, steps=64, ctrl-G=1.0, proj-G)

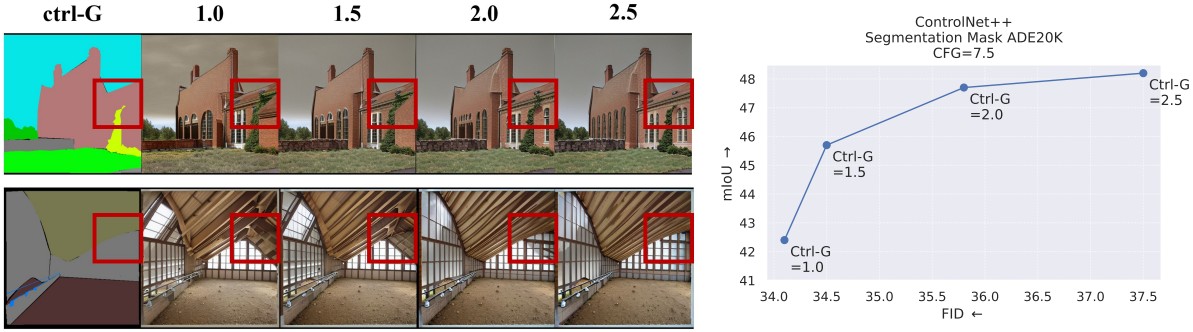

Figure 14: Ctrl-G can be applied to pre-trained opensource control adapters out of the box without the need for additional training. **Left**: Qualitative effect of increasing ctrl-G for ControlNet++ – consistency can be visually improved. **Right**: Quantitative tradeoff between consistency (segmentation mIoU) and quality (FID).

