# OpenReview forum: "A Practical Investigation of Spatially-Controlled Image Generation with Transformers"
_TMLR — Accepted by TMLR_

### Review · Reviewer_Y1Mu · 2025-08-08

**Summary Of Contributions:**

This paper proposes a study on what matters for spatially controlled image generation. The main contributions are: a new simple prefilling baseline that outperforms controlnet, a study on sampling and a study on the advantages and drawbacks of full fine-tuning versus frozen models. The study is based on a large set of experiments using the ImageNet image generation benchmark. Finally, the authors derive and discuss practical recommendations for researchers.

**Audience:**

Yes

**Audience Explanation:**

Strenghts:

- Despite the lack of global coherence described above, the paper presents interesting and useful findings for the community working on spatially-aware image generation. The 3 contributions: prefill baseline, the role of sampling and the usefulness of frozen settings against catastrophic forgetting are novel findings well supported by the experiments, and that could be applied to any generative model. In particular, the prefilling baseline outperforms controlnet, despite another paper finding it is bad, this paper shows that quantization was the issue and hurts the performance.

- The main findings are clearly highlighted by the presentation, which help grasp the contributions of the paper.

- The paper doesn’t try to sell something bigger than what is actually shown by the experiments. The reader can directly be convinced of the findings by looking at the experiments and can easily apply the findings to their own work.

**Claims And Evidence:**

Yes

**Claims Explanation:**

Strenghts:

- The experimental protocol based on ImageNet image generation, depth and canny controls, and quality + fidelity metrics is very clear and allows to make fair comparisons between a large variety of setups and models.

Weaknesses:
- Most of the tables are very complex, hard to read, and do not go straight to the point. Most tables are not self-contained and it is very hard to derive the findings of the paper just by looking at Tables. The names of the models are unnecessarily complex, for example “VAR-d16 CFG=2.5, temp=1, canny prefill + ctrl quant top-p=0.96, top-k=900”. The tables are large and a single Table is often used for several conclusions. Maybe think about how to improve the presentation and split the Tables ?

- The paper presents interesting findings when taken separately, but it feels like a list of independent experiments with no global coherence. What happens when you put all the improvements together: best sampling parameters with strongest model and adapter, how would this compare to current SOTA ?

- Figure 4 and 5 presents many settings with trade-offs between controllability and generation quality. These performance values are a bit abstract, what would you recommend concretely for practitioners aiming at improving qualitatively their generative models ? It feels that the discussion is missing the qualitative aspect.

- The design choices ablated in the paper (VAR and SIT, Sampling, Adapters) seem to be arbitrary: why ablating these components in particular ? The paper is lacking a discussion and some story telling on why these choices in particular, and about other interesting components that are not studied.

**Requested Changes:**

See weaknesses.

- Add global comparisons putting together all improvements
- Add missing recommandations for practitioners
- Add discussions on ablation design choices for global coherence and story telling
- Improve the clarity of the Tables

---

> ### Author Response · Authors · 2025-08-20
>
> We thank the reviewer for their feedback and are grateful that they appreciate our work.
>
> Here is our response to the reviewer's comments:
>
> `table clarity`
>
> We agree that the tables are difficult to parse, and have made the following improvements (Fig. 2 and Tab. 1. of the updated manuscript).
> 1. Annotations (arrows) linking takeaways in the captions to comparisons in the tables.
> 2. Reduced fontsize for sampling parameters (we agree that stating these adds clutter, but we believe it is important to explicity state them given they are varied in other places in the paper).
> 3. Clearly grouping columns into the type of metric they report (generation quality, control consistency, inference cost).
>
> Hopefully, this strikes a better balance between readability and providing the reader with complete results.
>
> `lack of global comparisons/putting together all improvements`
>
> The knowledge/experiments presented in this paper are not meant to point to a single optimal setting. Instead, the takeaways presented are each meant to separately inform the reader as to what suits their individual use case. We are not trying to say "this is the best way to do this"; rather, "here is information that is useful for performing this task". For example,
> > Takeaway: Control guidance greatly improves control consistency, but there is a trade-off against image quality and
> inference cost.  CFG has a small negative effect on consistency but generally improves generation quality.
>
> Informs the reader of the potential usefulness of control guidance and CFG; however, whether they use ctrl-G/CFG and how they set it depends on the reader's individual requirements on generation quality, consistency and computational cost. Thus, which takeaways to act on, and how to put them together should vary from practictioner to practictioner. **We have added a takeaway in Sec 3 to emphasise how  practitioners’ requirements subjectively depend on their individual use cases**.
>
> `missing qualitative recommendations`
>
> Figs. 4 and 5 demonstrate quantitative trends and trade-offs and Fig. 3 (and newly added Fig. 14) show qualitative effects that we expect to generalise beyond the specific models examined in this paper. To re-use the above example, Fig. 4 and the quoted takeaway would inform the reader of the general effects of ctrl-G/CFG, however, it depends on the reader's use case as to how they "turn these knobs". They may desire to meet certain numerical thresholds (e.g. <30 RMSE) or "eyeball" subjectively acceptable consistency (see newly added Fig. 14). Moreover, the specific values they arrive at will also depend on their own model.
>
> `arbitrary design choices`
>
> We direct the reviewer to the introduction, where we justify the investigative choices of the paper. Again, rather than aiming to build the "best" system (which is ill-defined) we are aiming to **clarify knowledge gaps** that we have identified in the literature. We understand that this may appear arbitrary; however, this is simply because such gaps naturally don't appear in a structured manner, and thus don't necessarily lend themselves to a nice clean story.
>
> The choice of VAR and SiT aims to best cover modern and prominent generation paradigms, given our constrained experimental resources.
>
> We again thank the reviewer for their service. We hope that the above response has helped clarify the nature of our paper. If you have further queries, please do not hesitate to let us know.

---

### Review · Reviewer_Ywws · 2025-08-14

**Summary Of Contributions:**

The paper systematically studies spatially-controlled image generation in terms of generation quality and consistency. Specifically, it studies effects of architecture, inference guidance scale and parameter-efficient fine-tuning method such as adapters. The paper provides empirical evidence to show that 1) pre-filing is a good conditioning mechanism for spatially-controlled image generation 2) separate guidance scale tuning for conditions can generally improve consistency and 3) adapters are good at maintaining image generation quality but weaker on consistency.

**Audience:**

Yes

**Audience Explanation:**

The paper studies image generation which fits TMLR's scope.

**Broader Impact Concerns:**

The paper doesn't have negative ethical implications.

**Claims And Evidence:**

Yes

**Claims Explanation:**

Strength:
- The paper conducts controlled experiments ablating various design choices. The empirical results are convincing and conclusions are intuitive.
- The paper studies both diffusion-based and autoregressive based generation paradigms, providing a more holistic view of effects of inference techniques such as guidance scale.
- The controlled experiments on 'forgetting' of full-finetuning is illustrative and shows the importance of curating balanced finetuning data.



Limitations:
- The paper's exploration is restricted to two types of controls, edge map and depth map.
- Guidance scale tuning for conditions has been explored in literature. While this exploration may not have been reported in literature, it is a common technique that people would tune in practice. This reduces the novelty of the takeaways of the paper.

Overall, the paper conducts good controlled experiments and nicely summarizes findings.

**Requested Changes:**

No major changes requested. Please see the Limitations.

---

> ### Author Response · Authors · 2025-08-20
>
> We thank the reviewer for their feedback and are grateful that they appreciate our work.
>
> Here is our response to the reviewer's comments:
>
> `only two types of control`
>
> We selected canny and depth maps as two distinct types of control that are similar to other popular control types. Canny maps provide fine-grained guidance on details and edges, like HED maps or line art, whilst depth maps provide smoother/coarser guidance across the whole image, similar to normal or segmentation maps.
>
> Besides, we have added an additional experiment using pre-trained ControlNet++ that demonstrates the effect of ctrl-G for segmentation maps as input control (newly added Appendix C, Fig. 14).
>
> `ctrl-G is commonly tuned in practice`
>
> We acknowledge the reviewer's point; however, we still believe it is valuable to present this knowledge clearly to the community as part of a referenceable and discoverable research paper.
>
> We again thank the reviewer for their service. If you have further queries, please do not hesitate to let us know.

---

### Review · Reviewer_SyWM · 2025-08-16

**Summary Of Contributions:**

This paper studies spatially controlled image generation with transformers, comparing various strategies used for this task across two different generative model architectures: a diffusion/flow model (SiT) and an autoregressive model (VAR). The authors explore approaches such as control token prefilling, tuning of sampling parameters (control guidance (ctrl-G), top-p sampling), and training two types of adapters for model weights. The main takeaways are the following:
-  prefilling is a strong, general baseline
- ctrl-G greatly improves control consistency, but there is a trade-off with image quality and
inference cost. It can be combined with other approaches for controllable image generation, such as adapters.
- top-p sampling can improve both generation quality and control consistency for VAR at inference time for no extra cost, but does not meaningfully benefit the consistency or generation quality of diffusion/flow models
- adapters mitigate forgetting in case of limited available data, but full model fine‑tuning performs better in terms of control–generation consistency.

Strengths:
- In my opinion, the topic explored in the paper (analysis of different techniques used for a single task, their comparison and applicability to different models) is highly important. We need more such work in different areas, especially in fields flooded with research papers such as generative modelling. It helps researchers better assess the value of new work, critically evaluate claims, and helps practitioners navigate tools for their needs
- Multiple simple strategies for spatially controlled image generation are considered. A simple yet strong and broadly applicable baseline is established (prefilling)
- The paper provides a thorough and transparent comparison of different approaches.
- The paper is well structured, with key takeaways clearly highlighted.
- The practical takeaways are valuable and will clearly help researchers establish solid baselines and practitioners tune model architectures and parameters for their needs.

 Weaknesses:
- The scope of models and datasets used is quite narrow. For models, only VAR-d16 and SiT-XL/2 are considered, both initialized with ImageNet-pretrained checkpoints. For data, all experiments use class-conditional ImageNet at a resolution of 256×256. Only Canny and DPT depth controls are considered. However, there is much more variety in control modes, such as text prompts, pose, segmentation, or normal maps. Moreover, modern image generators typically produce images at higher resolutions than 256×256. This may make the paper’s findings regarding general image quality less applicable to modern models, as well as its findings on the trade-off between control consistency and generation quality.
- FID/IS and pixel-level control metrics are used. It would be more convincing to include a user study or structure-aware/semantic adherence metrics (e.g., edge alignment tolerance, perceptual similarity). These metrics may better reflect downstream creative adequacy and would be extremely beneficial for practitioners. Moreover, combination of the dataset used (ImageNet) and metrics sometimes can lead to ambiguous results. For example, authors note: "We note that control-conditioned IS is lower than for generations without control; however this is to be expected since the control inputs are sourced from the ImageNet validation dataset, which naturally has lower IS/more ambiguous samples"
- Cross-paper comparisons seem somewhat fragile. While the authors reproduce or control much of the setup, some external baselines are quoted from prior work with potentially different training/data choices. Even the authors note that such comparisons can be unfair. For example: "We note that our result goes against the results in ControlAR (Li et al., 2025) that suggest that prefilling is a poor choice. We hypothesise that Li et al. (2025)’s choices to 1) vector quantise the control input, destroying information, and 2) enforce a triangular causal attention mask, limiting attention between control tokens, may have hurt performance". This relates to the earlier point about choice of models and datasets. It would be useful here to examine control-token attention patterns—comparing inter-control attention with triangular masks—to directly validate the hypothesis about why Li et al. (2025)’s design underperformed.
 - Strictly speaking, the latency accounting excludes some costs, as the authors themselves point out: “We do not include the decoder to the pixel space, or the extraction and encoding of control data to the latent space.” This may distort the overall picture of the benefits and drawbacks of different approaches for practitioners
- LoRA is quite a common approach for image generation control. It should therefore be considered alongside the other control methods. Including it would give the adapters section of the paper fuller coverage.

Some more suggestions around experiments discussed in the paper:
- In top-p experiment, It would be interesting to add per‑scale top‑p in VAR and contrast top‑k vs top‑p sweeps to map diversity/consistency trade‑offs more fully
- For the “forgetting mitigation” problem, it would be valuable to evaluate widely used mitigation strategies such as mixed-task fine-tuning (e.g., including some general text-to-image batches not related to the control setting during fine-tuning) to see whether they help reduce the problem

**Audience:**

Yes

**Audience Explanation:**

I do believe that the topic of the paper (study of approaches for spatially controlled image generation, with comparision of various strategies) if extremely important and many people would be interested in that. I also believe that the paper provides clear and valuable enough findings in this topic to be of interest for both researchers and practitioners.

**Broader Impact Concerns:**

-

**Claims And Evidence:**

Yes

**Claims Explanation:**

I do think the claims made in the submission are mostly supported by accurate, convincing and clear evidence. My main concerns regard that some experiments should be extended to build a better general picture around the topic of the paper (e.g. use different models, datasets, adding ablations on attention maps to understand why prefilling fails in ControlAR, include LoRA)

**Requested Changes:**

The main changes that I would strongly recommend are the following:
- Test results on a modern real-world model with real-world data used for fine-tuning (in the case of the adapter approach). This would help in the following ways: 1) Strengthen the paper’s claims by providing clear evidence of applicability in real-world scenarios and clarify how the choice of dataset (ImageNet) and models influenced the results. Since the paper claims that the work is intended to help practitioners, this is especially important. 2) Help explain the discrepancy between prefilling results and those reported in ControlAR.
- Include additional metrics to provide more convincing comparisons.
- Add at least one version of a LoRA adapter to the two adapter approaches discussed.

All other suggestions are mentioned in the “Summary Of Contributions” section. While they would be highly beneficial in my opinion, the three points above are the most crucial.

---

> ### Author Response · Authors · 2025-08-20
> **response (1/2)**
>
> We thank the reviewer for their feedback and are grateful that they appreciate our work.
>
> Here is our response to the reviewer's comments:
>
> `requested additional experiments`
>
> We would like to be transparent with the reviewer about this. The authors do not currently have the engineering or computational resources to run extensive additional experiments within the author-reviewer discussion period (e.g. train another adapter architecture for both SiT and VAR and evaluate over multiple sampling settings, or fine-tune/train large-scale T2I foundation models). We will offer some feasible suggestions later on that we can complete during the discussion window. This is due to the bulk of experiments in the paper being performed during an internship that has since concluded. We hope that, nevertheless, we can improve our manuscript during this discussion period to your satisfaction.
>
> `Choice of ImageNet 256 for experimentation`
>
> 1. We would like to re-iterate/clarify our motivation for using ImageNet as a basis for our experiments. Given the authors' modest engineering and computational resources as academic researchers, it allows us to control the data distribution for both *pre-training* and fine-tuning. This is not typically performed when comparing spatially controlled models in the literature (e.g. ControlAR) and is a confounding factor that we seek to remove.
> 2. We would like to politely contest the point that 256$\times$256 is too low a resolution and that ImageNet models/data are not "real world". Firstly, inspection by eye demonstrates that image semantics and details are well depicted at this resolution, and are similar to e.g. 512$\times$512. Secondly, ImageNet contains semantically diverse and complex images sourced from the real internet.  Finally, ImageNet 256 is the de facto benchmark in the literature for academic research on novel image generation models. For example, recently published work [1,2] use this benchmark, and do not report higher resolution or text-to-image results. Moreover, strong ImageNet performance for architectures often leads to successful text-to-image implementations ([3] for VAR, [4] for SiT).
>
> `Only Canny and DPT depth controls are considered`
>
> Given our resource constraints, we selected two representative controls that are similar to other popular control types. Canny maps provide fine-grained guidance on details and edges, like HED maps or line art, whilst depth maps provide smoother/coarser guidance across the whole image, similar to normal or segmentation maps.
>
> Nevertheless, **we acknowledge the above two concerns raised by the reviewer and have added an additional experiment**. Newly added Appendix C and Fig. 14 demonstrate the trade-off between control consistency and image quality when using ctrl-G with ControlNet++ on 512 resolution text-to-image data. It demonstrates generalisation of results to another condition (segmentation maps) as well as how ctrl-G can be leveraged out of the box for pretrained opensource control adapters, without any additional training or parameter overhead.
>
> [1] Wang et al, Parallelized Autoregressive Visual Generation *CVPR 2025*
>
> [2] Yao et al, Reconstruction vs. Generation: Taming Optimization Dilemma in Latent Diffusion Models *CVPR 2025*
>
> [3] Han et al. Infinity: Scaling Bitwise AutoRegressive Modeling for High-Resolution Image Synthesis *CVPR 2025*
>
> [4] Esser et al. Scaling Rectified Flow Transformers for High-Resolution Image Synthesis *ICML 2024*

---

> > ### Author Response · Authors · 2025-08-20
> > **response (2/2)**
> >
> > `only FID/IS and pixel-level control metrics`
> >
> > We use standard metrics adopted in the literature (the same as in ControlAR,ControlNet++) for evaluating adherence to control signals. To respond to the reviewer's suggestions
> > 1. F1 score, which we use, directly measures output edge alignment with the input condition
> > 2. It is unclear how one would easily measure semantic adherence between a control input (e.g. depth map) and a generated output image. A perceptual similarity loss like LPIPS would not be suitable.
> > 3. We agree that a user study would improve the evaluation, but we unfortunately do not have the means to complete one within the discussion window. We agree that the presented metrics only capture certain aspects of generation and that practitioners' requirements subjectively depend on their use case. We have added a takeaway in Sec. 3 reminding readers to interpret the results/metrics in the paper with this in mind.
> >
> > > ambiguous results
> >
> > We are unclear what the reviewer is referring to here. To clarify the quoted line relating to the ImageNet validation dataset: IS prefers higher confidence softmax scores from the inception network used for evaluation. Since the control source (ImageNet Val) contains images where the class is ambiguous, conditioning on these class-ambiguous controls will lead to generations that induce higher uncertainty softmax scores, reducing the inception score compared to a model without control.  Here the ambiguity refers to naturally occuring uncertainty in the data distribution. We have additionally clarified Inception Score in Sec. 3.
> >
> > `comparison with ControlAR is fragile`
> >
> > Results compared against from quoted papers are all on the ImageNet 256 benchmark (Fig. 2). The weak prefilling performance from ControlAR discuss is also on ImageNet 256, but it is not the quoted performance compared against (prefilling is a design choice that is discarded by the authors of ControlAR). The version of ControlAR compared against is their "best" imagenet 256 architecture and uses additive injection of control information, as such we believe the comparison is fair. We have added this clarification to the paper.
> >
> > We would also like to add that Sec. 4 aims to establish prefilling as a simple and effective *baseline*, which we can then use as a basis for further experiments, not to demonstrate that it is the most superior choice or to thoroughly explain the behaviour of ControlAR.
> >
> > `latency accounting excludes some costs`
> >
> > We acknowledge the reviewer's point, however, we believe that the *relative* comparisons and takeaways in the paper are still valid, since the excluded computations are fixed and independent of the design choices that we are comparing between. Moreover, we believe that measuring only the generative component illustrates compute-scaling behaviour (e.g. add adapter approximately doubles cost) that is more generalisable beyond the experiments in this work (which conversely including the decoder may obscure).
> >
> > `absence of LoRA experiments`
> >
> > The takeaways presented in the paper are separate to LoRA, however, we agree with the reviewer that LoRA is a relevant approach.  As such LoRA experiments would add *additional scope* to the paper.
> > We would like to clarify their request to
> > > Add at least one version of a LoRA adapter to the two adapter approaches discussed.
> >
> > We would like to check if the reviewer means to take e.g. the ControlNet adapter and only train a LoRA attached to the ControlNet? The suggested additional experiment would add an *additional takeaway* to the paper along the lines of:
> > > Takeaway: Using LoRA on the trainable copy improves parameter efficiency at the cost of control consistency.
> >
> > On the other hand, prefilling finetuning could be performed directly with LoRA, which would instead contribute to the existing takeaway
> > > Takeaway: Adapters, where the generative model is frozen, can learn spatial conditioning. Control consistency is consistently worse than prefill + finetuning all parameters, although good generation quality is maintained.
> >
> > We would likely be able to engineer, train and evaluate one LoRA model on one control type during the discussion period, so we would like to know the reviewer's preference.
> >
> >
> > We again thank the reviewer for their service, and hope to take to time to consider our response. We apologise that we may not have the means to fully implement your requests within the discussion period and hope for your understanding on this matter. If you have further queries, please do not hesitate to let us know.

---

> ### Author Response · Authors · 2025-08-28
> **further update**
>
> We have added a supplementary ablation and updated the manuscript (see Appendix A2 Table 2). We apply LoRA both to prefilling as well as the prefill adapter (i.e. replacing fully trainable parameters in the Figure 6 illustration of different architecture approaches with LoRA adapters).
> It demonstrates that LoRA enables spatial controllability with greater parameter efficiency at the cost of generation and consistency performance.
>
> We hope that this is to your satisfaction.

---

### Author Response · Authors · 2025-08-20
**General Response**

We would like to thank the reviewers for their service.

We are grateful that they find:
- the explored research topic "highly important"
- our empirical comparisons "transparent", "thorough" and "fair",  and our results and conclusions "convincing" and "intuitive"
- our practical takeaways/findings to be "valuable", "interesting" and "useful for the community"
- the presentation "well structured", with "key takeaways/main findings clearly highlighted"

**We will briefly summarise updates to the manuscript**:
1. We have clarified FID/IS in Sec 3. and emphasised that practitioners' requirements subjectively depend on their use case.
1. The tables in Fig. 2 and Tab. 1 have been improved for better readability. Columns are now explicitly grouped by metric type, and results described in the caption are annotated on the table, directing the reader's eyes to the relevant numbers.
2. We include an additional experiment (new Appendix C, Fig. 14) demonstrating that ctrl-G can be applied out of the box to a pre-trained text-to-image control adapter (ControlNet++), trading off FID for better control consistency.

---

> ### Author Response · Authors · 2025-08-28
> **Additional update**
>
> Dear all reviewers,
>
> In response to reviewer SyWM's request for LoRA experiments, we have added a supplementary ablation (see Appendix A2 Table 2). It demonstrates that LoRA enables spatial control with greater parameter efficiency at the cost of generation and consistency performance.

---

### Decision · Action_Editor_4ccN · 2025-10-07

**Recommendation:** Accept with minor revision

**Audience:**

Yes

**Audience Explanation:**

As detailed above, the authors conducted well-controlled experiments that yield clear and interpretable takeaways. The key findings, such as prefilling serves as a simple yet strong baseline, are valuable contributions to the community of spatially controlled image generation.

**Claims And Evidence:**

Yes

**Claims Explanation:**

The diversity in experimental settings (e.g., training data and architectures) in spatially-controlled image generation makes it hard to isolate the sources of key improvements. In this work, the authors conduct systematic and controlled experiments on ImageNet to provide clear takeways across different generation paradigms. The key findings include: (1) control-token-prefilling serves as a simple, general, and strong baseline; (2) a careful analysis of sampling time adjustments enhances control-generation consistency; and (3) a comparative study highlights the advantages and limitations of adapter-based vs. full fine-tuning approaches. The claims are empirically validated using two types of control signals (edge and depth maps) and two generative models (SiT and VAR).

Initially, all three reviewers acknowledged that the paper provides practical contributions to the community of spatially controlled image generation. However, they also raised several concerns, including the need for improved writing quality and limitations in the experimental setup. After the rebuttal, most of these concerns were satisfactorily addressed.

In the final recommendation stage, two reviewers leaned toward acceptance, while one leaned toward rejection, primarily citing issues with the experimental setup. Specifically, the reviewer expressed concerns about the narrow scope of the dataset (ImageNet-256), models (limited to SiT and VAR), and control signals (restricted to edge and depth maps).

After carefully reviewing the submission, the reviews, and the author rebuttal, the action editor acknowledges the remaining concerns but appreciates the practical value and findings presented in the paper. To strengthen the claims and ensure proper contextualization, the action editor recommends "*Accept with minor revision*", suggesting that the authors include a “Limitations” section to explicitly discuss the constraints of the current experimental setup. This addition will help better support and contextualize the paper’s claims.